

# Multi-model study of mercury dispersion in the atmosphere: Vertical distribution of mercury species

Johanes Bieser[1,2], Franz Slemr[3], Jesse Ambrose[4], Carl Brenningkmeijer[3], Steve
Brooks[5,6], Ashu Dastoor[7], Francesco DeSimone[8], Ralf Ebinghaus[1], Christian N.
Gencarelli[8], Beate Geyer[1], Lynne E. Gratz[9], Ian M. Hedgecock[8], Daniel Jaffe[4,10], Paul
Kelley[5,11], Che-Jen Lin[12], Volker Matthias[1], Andrei Ryjkov[7], Noelle E. Selin[13,14],
Shaojie Song[13], Oleg Travnikov[15], Andreas Weigelt[1,16], Winston Luke[5], Xinrong
Ren[5,11,17], Andreas Zahn[18], Xin Yang[19], Yun Zhu[20], Nicola Pirrone[21]

1    Helmholtz Zentrum Geesthacht, 21052 Geesthacht, Germany

2    DLR – Deutsches Luft und Raumfahrtzentrum, Münchener Straße 20, 82234 Weßling, Germany

3    Max-Planck-Institute for Chemistry (MPI), Hahn-Meitner-Weg 1, 55128 Mainz, Germany

4    School of Science, Technology, Engineering and Mathematics, University of Washington-Bothell, Bothell, WA, USA

5    Air Resources Laboratory, National Oceanic and Atmospheric Administration, 5830 University Research Court, College Park,
MD 20740, USA

6    Department of Mechanical, Aerospace and Biomedical Engineering, University of Tennessee Space Institute, 411 BH Goethert
Parkway, Tullahoma, TN 37388, USA

7    ECCC – Air Quality Research Division, Environment and Climate Change Canada, Dorval, Canada

8    CNR-Institute of Atmospheric Pollution Research, Division of Rende, Rende, Italy

9    Environmental Program, Colorado College, Colorado Springs, CO, USA

10    School of Science, Technology, Engineering and Mathematics, University of Washington-Bothell, Bothell, WA, USA

11    Cooperative Institute for Climate and Satellites, University of Maryland, 5825 University Research Court, College Park, MD
20740, USA

12    Center for Advances in Water and Air Quality, Lamar University, Beaumont, Texas, USA.

13    Department of Earth, Atmospheric and Planetary Sciences, Massachusetts Institute of Technology, Cambridge, MA, USA

14    Institute for Data, Systems, and Society, Massachusetts Institute of Technology, Cambridge, MA, USA

15    Meteorological Synthesizing Centre – East of EMEP, Moscow, Russia

16    Federal Maritime and Hydrographic Agency (BSH), Hamburg, Germany

17    Department of Earth, Ocean, and Atmospheric Science, Florida State University, 117 North Woodward Avenue, Tallahassee, FL
32306, USA

18    Institut für Meteorologie und Klimaforschung (IMK-ASF), Karlsruhe Institut für Technologie, Hermann-von-Helmholtz-Platz 1,
76344 Leopoldshafen, Germany

19    British Antarctic Survey, Cambridge, UK

20    South China University of Technology, School of Environment and Energy, Guangzhou, China

21    CNR Institute of Atmospheric Pollution Research, Rome, Italy





Abstract

Atmospheric chemistry and transport of mercury play a key role in the global mercury cycle. However, there are still considerable knowledge gaps concerning the fate of mercury in the atmosphere. This is the second part of a model inter-comparison study investigating the impact of atmospheric chemistry and emissions on mercury in the atmosphere. While the first study focused on ground based observations of mercury concentration and deposition, here we investigate the vertical distribution and speciation of mercury from the planetary boundary layer to the lower stratosphere. So far, there have been few model studies investigating the vertical distribution of mercury, mostly focusing on single aircraft campaigns. Here, we present a first comprehensive analysis based on various aircraft observations in Europe, North America, and on inter-continental flights.

The investigated models proved to be able to reproduce the distribution of total and elemental mercury concentrations in the troposphere including inter-hemispheric trends. One key aspect of the study is the investigation of mercury oxidation in the troposphere. We found that different chemistry schemes were better at reproducing observed oxidized mercury (RM) patterns depending on altitude. High RM concentrations in the upper troposphere could be reproduced with oxidation by bromine while elevated concentrations in the lower troposphere were better reproduced by OH and ozone chemistry. However, the results were not always conclusive as the physical and chemical parameterizations in the chemistry transport models also proved to have a substantial impact on model results.



## 1. Introduction

At the time of writing the Minamata Convention has 128 signatories and has been ratified by 35 countries.

This international legally binding treaty will oblige all participating parties to:

I)     Assess the state of mercury pollution

II)    Take actions to reduce mercury emissions and concentrations in the environment

III)   Evaluate the success of the measures taken on a regular basis.


The state of mercury contamination is typically determined by measurement of the relevant mercury species (e.g. total mercury in the atmosphere, methylmercury in fish). However, in order to understand the sources of mercury pollution and to predict the impact of various possible measures for mercury

emission reduction it is necessary to apply complex chemistry transport models. In the last decades, general chemistry transport models (CTMs) have been extended to model the global mercury cycle by including mercury chemistry and partitioning (Bergan et al., 1999; Xu et al., 2000; Lee et al., 2001; Peterson et al., 2001, Seigneur et al., 2001; Dastoor et al., 2004; Selin et al., 2007). Since then,

extensive model inter-comparison studies have been performed to evaluate and improve the original models (Bullock et al., 2008; Ryaboshapko et al., 2007). However, up until today, we have not fully understood all parts of the global mercury cycle. In the atmosphere, the main question is how elemental mercury emitted from anthropogenic, natural, and legacy sources and then oxidized. This

includes the relative importance of oxidizing reaction partners and the possibility of reduction pathways of oxidized mercury. Once we understand the red-ox processes of atmospheric mercury, is it possible to determine the range of mercury transport and the fate of mercury emitted in the past and the future. Consequently, mercury oxidation processes have been in the focus of the

international mercury community in recent years (Cohen et al., 2016; Amos et al., 2015; Dastoor et al., 2015; Song et al., 2015; Bieser et al., 2014; De Simone et al.,





2014; Qureshi et al., 2011; Travnikov et al., 2010).

In this study, we investigate the vertical distribution of mercury species in the atmosphere. While gaseous elemental mercury (GEM) makes up the vast majority

of total atmospheric mercury near the surface (Sprovieri et al., 2016 this issue), recent aircraft based observations have indicated that there is significant oxidation of mercury occurring in the free troposphere (Brooks et al., 2014; Jaffe and Lyman, 2012; Jaffe et al., 2014; Gratz et al., 2014; Shah et al., 2015). As gaseous oxidized mercury (GOM) is much more rapidly removed from the

atmosphere than elemental mercury the free troposphere, the region between the planetary boundary layer and the tropopause is of great importance for the global mercury budget.

To investigate this issue further, the Mercury Modelling Task Force (MMTF) was founded during the course of the EU FP7 project GMOS (Global Mercury

Observation System). The MMTF is a global collaboration, not limited to GMOS project partners and thus, incorporates most mercury CTMs currently in use in the scientific community. With a total of seven model combinations (including four global, one hemispheric, and two regional models), the partners in the MMTF carried out a set of sensitivity model runs and compared the results to airborne

observations in Europe, North America, and on intercontinental flights.

## 2. Methods

### 2.1 Observations

Aircraft based observations are expensive and thus rarely performed on a regular basis. They are made in a certain area at a limited time interval and as such are hardly representative enough to be used to evaluate model performance. However, in the year 2013 an unprecedented amount of aircraft based observations has been made:

Within the European Tropospheric Mercury Experiment (ETMEP) 5 vertical profiles were flown in the planetary boundary layer (PBL) and the lower free troposphere (LFT) at an altitude of 500 – 3500m over central Europe during



August 2013 (Weigelt et al., 2016a). Mercury was measured using two Tekran instruments (2537X and 2537B). Both Tekran instruments were run with upstream

particle filters and one, additionally, with a quartz wool trap which presumably removes gaseous oxidized mercury (GOM) (Lyman and Jaffe, 2011; Ambrose et al., 2013). Neglecting PBM the concentrations of which is usually negligible, the measurement by Tekran without the quartz wool trap approximates TM and that with quartz wool trap GEM (Weigelt et al., 2016b). GEM was also measured by a

modified Lumex instrument (Weigelt et al., 2016b). Additionally, gaseous oxidized mercury (GOM) was collected on denuders and analyzed on return to the laboratory.

In the U.S. Brooks et al. (2014) measured GEM, GOM, and PBM (particulate bound mercury) profiles on 28 flights between August 2012 and July 2013 at

altitudes from 1000m to 6000m. GEM was measured on board with a modified Tekran 2537B instrument with a temporal resolution of 2.5 minutes. GOM was collected on denuders and PBM on a filter tube downstream of the denuder. Both were later analyzed in the laboratory. In addition, 19 flights were flown in June and July 2013 mostly over the south-eastern USA  at altitudes between 500m – 7000m

during the NOMADSS (Nitrogen, Oxidants, Mercury and Aerosol Distributions, Sources and Sinks) campaign (Gratz et al., 2015; Shah et al., 2016). Here, oxidized mercury was calculated based on a differential method using two Tekran 2537B instruments, one of which was equipped with GOM trap (quartz woll or ion-exchange membrane (DOHGS) (Lyman and Jaffe, 2011; Ambrose et al., 2015).

Finally, there were 19 intercontinental flights between Germany and North and South America were made within the CARIBIC (Civil Aircraft for the Regular Investigation of the atmosphere Based on an Instrumented Container) project during which TM and GEM was measured in the upper troposphere and the lower stratosphere in altitudes between 6000m – 12000m using a modified Tekran 2537

A instrument (Slemr et al., 2014; 2016).

The aircraft observations were complemented with ground based observations from the GMOS measurement network (Sprovieri et al., 2016; GMOS, 2016). In particular, we used data from the ground based stations in Mace Head, Ireland





and Waldhof, Germany to augment the ETMEP profiles (Weigelt et al., 2013;
2014). At Mace Head and Waldhof GEM is measured with a Tekran 2537 A. At
Waldhof, additionally, GOM and PBM are measured with a Tekran 1130/1135
speciation unit.

## 2.2 Models

This study is based on an ensemble of seven different CTMs including global
(GLEMOS, GEOS-Chem, GEM-MACH-Hg, ECHMERIT), hemispheric (CMAQ-Hem),
and regional (WRF-Chem, CCLM-CMAQ) models (Table 1). The models differ
considerably in the implemented physical and chemical parameterizations, spatial
and temporal resolution, and meteorological drivers. The ensemble includes
models that use external fields for chemical reaction partners (GLEMOS, GEOS-
Chem), models with a complete photochemical reaction scheme (CCLM-CMAQ,
CMAQ-Hem) and on-line coupled meteorological models (GEM-MACH-Hg,
ECHMERIT, WRF-Chem). The only model harmonization in this study is the
utilization of a common global 1°x1° anthropogenic emission inventory
(AMAP/UNEP, 2013a; 2013b). However, the models use different temporal
disaggregation and down-scaling methods, source heights, and speciation
schemes to convert the global emission dataset into model ready input fields. The
main analysis of the vertical mercury distribution was performed using the
standard setup of each model (BASE case). The chemical mechanisms for mercury
oxidation in the BASE case can be grouped into three major classes:

1) Ozone and OH chemistry (GLEMOS, ECHMERIT, CMAQ-Hem, CCLM-
   CMAQ, WRF-Chem)
2) OH and bromine chemistry (GEM-MACH-Hg)
3) Bromine chemistry (GEOS-Chem)

Moreover, some models also consider reduction of $Hg^{2+}$ to GEM in the aqueous
phase (GLEMOS, ECHMERIT, WRF-Chem, CMAQ). In addition to the BASE cases, a
set of chemistry and emission sensitivity runs was performed. These include runs





with no anthropogenic emissions (NOANT) and with a 100% GEMspeciation of anthropogenic emissions (ANTSPEC). For the mercury chemistry, different runs with only one of the above mentioned oxidants (OHCHEM, O3CHEM, BRCHEM) and without any mercury chemistry (NOCHEM) were performed. Concerning the bromine reaction, two different Br and BrO fields were used. These are bromine

fields from GEOS-Chem (Parella et al., 2012) and the p-TOMCAT model (Yang et al., 2005, 2010). However, the described sensitivity runs were not performed by all models. Moreover, the list differs from that published by Travnikov et al. (2016, this issue) as only a limited set of 3D model output data could be saved. A synthetic model description is given in Table 1 and the sensitivity runs performed

are further described in Table 2. An evaluation of ground based mercury concentrations and deposition fluxes for the four global models (GLEMOS, GEOS-Chem, GEM-MACH-Hg, ECHMERIT) can be found in Travnikov et al. (2016, this issue). An evaluation of regional deposition fields can be found in Gencarelli et al. (2016, this issue). For the sake of completeness we provide the detailed model

descriptions here as well.

### 2.2.1 GLEMOS

GLEMOS (Global EMEP Multi-media Modelling System) is a multi-scale chemistry transport model developed for the simulation of environmental dispersion and

cycling of different chemicals including mercury based on the older hemispheric model MSCE-HM-Hem (Travnikov, 2005; Travnikov and Ilyin, 2009; Travnikov et al., 2009). The model simulates atmospheric transport, chemical transformations and deposition of three Hg species (GEM, GOM and PBM). The atmospheric transport of the tracers is driven by meteorological fields generated with the Weather

Research and Forecast modelling system (WRF 3.7.2) (Skamarock et al., 2007) which is fed by operational analysis data from the European Centre for Medium-Range Weather Forecast (ECMWF) (ECMWF, 2016). In the base configuration the model grid has a horizontal resolution of 1°×1°. Vertically, the model domain reaches up to 10 hPa and consists of 20 irregular terrain-following sigma layers.

The atmospheric chemical scheme includes Hg oxidation and reduction reactions




in both the gas phase and the aqueous phase of cloud water. The major chemical mechanisms in the gas phase include Hg oxidation by $O_3$ and OH radicals with reaction rate constants taken from Hall (1995) and Sommar et al. (2001), respectively. The latter was scaled down by a factor of 0.1 within and below clouds

to account for reduced photochemical activity (Seigneur et al., 2001). The $O_3$ and OH concentration fields are imported from MOZART (Emmons et al., 2010). A two-step gas-phase oxidation of GEM by Br is included as an option. Aqueous-phase reactions include oxidation by ozone, chlorine and hydroxyl radical and reduction via decomposition of sulphite complexes (Van Loon et al. 2000). The model

distinguishes in-cloud and sub-cloud wet deposition of PBM and GOM based on empirical data. The dry deposition scheme is based on the resistance analogy approach (Wesely and Hicks, 2000). Prescribed fluxes of natural and secondary emissions of Hg from soil and seawater were generated depending on Hg concentrations in soil, soil temperature and solar radiation for emissions from land

and proportional to the primary production of organic carbon in seawater for emissions from the ocean (Travnikov and Ilyin, 2009). In addition, an empirical parametrization of the prompt Hg re-emission from snow- and ice-covered surfaces is applied based on observational data.

2.2.2 GEOS-Chem

        The GEOS-Chem global chemistry transport model (v9-02; www.geos-chem.org) is driven by assimilated meteorological data from the NASA GMAO Goddard Earth Observing System (Bey et al., 2001). The GEOS-FP and GEOS-5.2.0 data are used for the simulation year 2013 and the spin-up period, respectively

(http://gmao.gsfc.nasa.gov/products/). GEOS-Chem couples a 3-D atmosphere (Holmes et al., 2010), a 2-D mixed layer slab ocean (Soerensen et al., 2010), and a 2-D terrestrial reservoir (Selin et al., 2008) in a horizontal resolution of 2°×2.5°. Three mercury species (GEM, GOM, and PBM) are tracked in the atmosphere (Amos et al., 2012). A two-step gaseous oxidation mechanism initialized by Br

atoms is used. Bromine fields are archived from a full-chemistry GEOS-Chem simulation (Parrella et al., 2012) while the rate constants of reactions are from





Goodsite et al. (2012), Donohoue et al. (2006), and Balabanov et al. (2005). The surface fluxes of GEM include anthropogenic sources, biomass burning, geogenic activities, as well as the bidirectional fluxes in the atmosphere-terrestrial and

atmosphere-ocean exchanges (Song et al., 2015). Biomass burning emissions are estimated using a global CO emission database and a volume ratio of Hg/CO of $1\times10^{-7}$. Geogenic activities are spatially distributed based on the locations of mercury mines. For atmosphere-terrestrial exchange, GEOS-Chem treats the evasion and dry deposition of GEM separately (Selin et al., 2008). Dry deposition

is parametrized with a resistance-in-series scheme (Wesely, 1989). In addition, an effective GOM uptake by sea-salt aerosol is also included over the ocean (Holmes et al., 2010). GEM evasion includes volatilization from soil and rapid recycling of newly deposited Hg. The former is estimated as a function of soil Hg content and solar radiation. The latter is modeled by recycling a fraction of wet/dry deposited

oxidized mercury to the atmosphere as GEM immediately after deposition (60% for snow covered land and 20% for all other land uses) (Selin et al., 2008). GEOS-Chem estimates the atmosphere-ocean exchange of GEM using a standard two-layer diffusion model. The ocean mercury in the mixed layer interacts not only with the atmospheric boundary layer but also with subsurface waters through

entrainment/detrainment of the mixed layer and wind-driven Ekman pumping (Soerensen et al., 2010).

### 2.2.3 GEM-MACH-Hg

GEM-MACH-Hg is a new chemical transport model for mercury that is based on the

GRAHM model developed by Environment and Climate Change Canada (Dastoor et al., 2004; 2008; 2010; Durnford et al., 2010; 2012; Kos et al., 2013) GEM-MACH-Hg uses a newer version of the Environment and Climate Change Canada's operational meteorological model. The horizontal resolution of the model is 1°×1°. GEM is oxidized in the atmosphere by OH radicals. The rate constant of the

reaction is from Sommar et al. (2001), but scaled down by a coefficient of 0.34 to take into account possible dissociation/reduction reactions (Tossell et al., 2003; Goodsite et al., 2004). The gaseous oxidation of mercury by bromine is applied in



polar regions using reaction rate constants from Donohoue et al. (2006), Dibble et al. (2012) and Goodsite et al. (2004). The parametrization of atmospheric mercury depletion events is based on Br production and chemistry, and snow re-emission of GEM (Dastoor et al., 2008).

OH fields are from MOZART (Emmons et al., 2010) while BrO is derived from 2007-2009 satellite observations of BrO vertical columns. The associated Br concentration is then calculated from photochemical steady state conditions (Platt and Janssen, 1995). Dry deposition in GEM-MACH-Hg is based on the resistance approach (Zhang, 2001; Zhang et al., 2003). In the wet deposition scheme, GEM and GOM are partitioned between cloud droplets and air using a temperature-dependent Henry's law constant. Total global emissions from natural sources and re-emissions of previously deposited Hg (from land and oceans) in GEM-MACH-Hg are based on the global Hg budgets by Gbor et al. (2007), Shetty et al. (2008) and Mason (2009). Land-based natural emissions are spatially distributed according to the natural enrichment of Hg. Terrestrial re-emissions are spatially distributed according to the historic deposition of Hg and land-use type and depend on solar radiation and the leaf area index. Oceanic emissions depend on the distributions of primary production and atmospheric deposition.

### 2.2.4 ECHMERIT

ECHMERIT is a global on-line meteorological chemistry transport model, based on the ECHAM5 global circulation model, with a highly flexible chemistry mechanism designed to facilitate the investigation of atmospheric mercury chemistry (Jung et al., 2009; De Simone et al., 2014, 2015, 2016). The model uses the same spectral grid as ECHAM. The standard horizontal resolution of the model is T42 (approximately, 2.8°×2.8°), whereas in the vertical the model is discretized with a hybrid-sigma pressure system with 19 non-equidistant levels up to 10 hPa. The base chemical mechanism includes the GEM oxidation by OH and $O_3$ in the gaseous and aqueous phases. Reaction rate constants are from Sommar et al. (2001), Hall (1995), and Munthe (1992), respectively. OH and $O_3$ concentration fields were imported from MOZART (Emmons et al., 2010). The Hg oxidation by Br





is also optionally available in a two-step gas phase oxidation mechanism with
reaction rates as described in Goodsite et al. (2004), Goodsite et al. (2012) and
Donohoue et al. (2006). ECHMERIT uses a parametrization of dynamic air-
seawater exchange as a function of ambient parameters, but using a constant
value of mercury concentration in seawater (De Simone et al., 2014). Emissions
from soils and vegetation were calculated off-line and derived from the
EDGAR/POET emission inventory (Granier et al., 2005; Peters and Olivier, 2003)
that    includes    biogenic    emissions    from    the    GEIA    inventories
(http://www.geiacenter.org), as described by Jung et al. (2009). Prompt re-
emission of a fixed fraction (20%) of wet and dry deposited mercury is applied in
the model to account for reduction and evasion processes which govern mercury
short-term cycling between the atmosphere and terrestrial reservoirs (Selin et al.,
2008). This fraction is increased to 60% for snow-covered land and ice covered
seas.

### 2.2.5 CMAQ-Hem

This is a hemispheric set-up of the Community Multi-Scale Air Quality System
(CMAQ) version 4.6 (Byun and Schere, 2006; Byun and Ching, 1999). The model is
based on a three-dimensional Eulerian atmospheric chemistry and transport
modeling system that simulates Hg, ozone, particulate matter, acid deposition,
and visibility simultaneously. The model components and scientific backgrounds
have been documented elsewhere (Bullock and Brehme, 2002; Bullock et al.,
2008; Travnikov et al., 2010). A spin-up period of 10 days is used to eliminate the
impact of initial conditions for atmospheric oxidants ($O_3$ and OH) that react with
mercury. As for mercury species, global models were simulated for several years
prior to the study period (2005) in order to provide the initial and boundary
conditions  for this study (Pongprueksa et al., 2011). A hemispheric model domain
with a Polar Stereographic projection at 108-km spatial resolution and 187 ×187
grid cells was used for this experiment with 13 sigma hybrid layers up to 50 hPa.
Hourly meteorological data were prepared using the Weather Research and
Forecasting (WRF) model Version 3.7 (Skamarock et al., 2008). The selected



physics options were Thompson (Microphysics Options) (Thompson et al., 2004),
Betts-Miller-Janjic (Cumulus Parameterization Options) (Janjic, 1994; 2000), RRTMG
(Radiation Physics Options) and BouLac (PBL Physics Options) based on the results
of meteorological model performance evaluation (Wang et al., 2014). The ARW
outputs were processed using MCIPv3.4.1 (Byun and Ching, 1999; Otte and Pleim,
2010) to generate model-ready meteorology for chemical transport simulations.

### 2.2.6 WRF-CHEM

The WRF/Chem-Hg model (Gencarelli et al., 2014; 2015; 2016) is a modified
version of WRF/Chem (version 3.4, Grell et al., 2005) model, developed to
reproduce the emission, transport, chemical transformation and deposition of Hg
at local scales with elevated spatial and temporal resolutions. The gas phase
chemistry of Hg and a parametrized representation of atmospheric Hg aqueous
chemistry have been added to the RADM2 chemical mechanism using KPP (Sandu
and Sander, 2006) and the WKC coupler (Salzmann and Lawrence, 2006), in order
to represent four Hg species: GEM, GOM, PBM, and dissolved oxidized mercury
($Hg^{II}_{(aq)}$) (see Gencarelli et al., 2014 for further details regarding Hg
parametrizations and the physics options employed). Oxidation by $O_3$, OH and Br
was implemented as described in Gencarelli et al., 2015, in accordance with the
experimental purpose. In the BASE case only $O_3$ and OH chemistry are used.
Chemical Initial and Boundary Conditions (IC/BC) were taken from the ECHMERIT
model (Jung et al., 2009; De Simone et al., 2014) for Hg species, while boundary
conditions for other chemical species were taken from MOZART-4 (Emmons et al.,
2010). Dry deposition of gas-phase species is treated using the approach
developed by Wesely (1989), multiplying the concentrations in the lowest model
layer by the spatially and temporally varying deposition velocity, which is
proportional to aerodynamic, sublayer, and surface resistances. The wet
deposition of Hg species has been implemented by adding the Hg compounds to
the scheme in WRF/Chem for gas and particulate convective transport and wet
deposition. In-cloud and below-cloud scavenging of Hg species have been treated
in accordance with the approach described by Neu and Prather (2012), with Hg



species scavenging rate assumed to be the same as that for $HNO_3$(g). The model domain covers Europe and the Mediterranean Sea, including part of the western North Atlantic Ocean, North Africa and the Middle East with a horizontal resolution of 24 × 24 km, and 30 vertical levels from soil to 50 hPa. Hg emissions by

AMAP/UNEP (2013a, 2013b) for mercury and from the EDGARv4.tox1 (2008) inventory for other species were interpolated on this model domain.

### 2.2.7 CCLM-CMAQ

This modelling system is based on the meteorological model CCLM and the

chemistry transport model CMAQ v5.0.1. All physical atmospheric parameters were taken from regional atmospheric simulations with the COSMO-CLM v4.8 mesoscale meteorological model (Geyer, 2014) using NCEP reanalysis data as forcing (Kalnay et al., 1996). COSMO-CLM is the climate version of the regional scale meteorological community model COSMO (Rockel et al., 2008), originally

developed by Deutscher Wetterdienst (DWD) (Steppeler et al., 2003; Schaettler et al. 2008). It has been run on a 0.22° x 0.22° grid using 40 vertical layers up to 20 hPa for the whole of Europe. COSMO-CLM uses the TERRA-ML land surface model (Schrodin and Heise, 2001), a TKE closure scheme for the planetary boundary layer (Doms, 2011; Doms et al., 2011), cloud microphysics after Seifert and

Beheng (2001, 2006), the Tiedtke scheme (Tiedtke, 1989) for cumulus clouds and a long wave radiation scheme following Ritter and Geleyn (1992). The meteorological fields were then processed to match the Lambert Conformal Conical CMAQ grid with a grid size of 24 x 24 km with 30 sigma hybrid layers up to 50 hPa. CMAQ uses the information that is provided by the meteorological input

fields to calculate transport, transformation and loss of all gas phase and particulate species (Byun & Ching, 1999; Byun & Schere, 2006). For this study we used the multi-pollutant version with the carbon bond 5 photochemical mechanism cb05tump (Tanaka et al., 2003; Yarwood et al., 2005; Sarwar et al., 2007; Whitten et al., 2010) and the aerosol module aero6 (Appel et al., 2013;

Carlton et al., 2010; Foley et al., 2010). Deposition schemes are based on Byun and Schere (2006) for dry and Pleim and Ran (2011) for wet deposition. The



mercury chemistry is based on Bullock and Brehme (2002) and was updated based on observations and model inter-comparisons in the course of the EU FP7 project GMOS (Global Mercury Observation System) (Zhu et al., 2015; Bieser et al., 2014a, 2014b). To describe the re-emission of deposited mercury we used the bi-directional flux parametrization following Bash et al. (2010). Additionally, emissions from the North- and Baltic Sea were estimated based on Bieser and Schrum (2016). Boundary conditions were obtained from the GLEMOS model for GEM, GOM, PBM (Travnikov and Ilyin, 2009) and from TM-5 for all other species (Huijnen et al., 2010). The annual total emissions are based on AMAP for mercury (AMAP, 2013a, 2013b) and EMEP for other species and were speciated and disaggregated to an hourly resolution with the SMOKE for Europe emission model (Bieser et al., 2011a). Plume rise of point sources was explicitly calculated based on Bieser et al., (2011b). Finally biogenic emissions were calculated on-line using the BEIS3.14 model (Schwede et al., 2005; Vukovich et al., 2002).

## 2.3 Sensitivity runs

To evaluate the impact of emissions and atmospheric chemistry on the vertical distribution of mercury a set of sensitivity runs was made. While for the BASE case each model uses it's default setup, for the sensitivity runs certain aspects of the models were harmonized. The list of all sensitivity runs is given in Table 2. Concerning emissions, we tested the impact of anthropogenic emissions by considering only natural and legacy emissions (NOANT) and by altering the speciation of anthropogenic emissions to 100% GEM (ANTSPEC). In addition, we investigated different oxidation reactions by considering only one reaction at a time, namely ozone (O3CHEM), hydroxy radicals (OHCHEM), and bromine (BRCHEM). In these cases, the models used the same input fields for the investigated reactant. For bromine chemistry two alternative sets of bromine fields were used from GEOS-Chem (BRCHEM1) and from the p-TOMCAT model (BRCHEM2).

## 2.4 Model evaluation





For the model evaluation the grid cell and time step matching each individual measurement were taken. This means for example that observations within a single vertical profile can correspond to different time steps in the model. Due to the small amount of aircraft observations available, such a comparison faces the problem that the model bias will not average out as it tends to do for larger data sets (e.g. 8760 hourly observations for a single year of ground-based station data). Moreover, the vertical model performance is highly dependent on meteorological parameters (e.g. PBL height, vertical transport). Thus, for an individual profile the model bias can be quite large. We did not perform a detailed analysis of the meteorological fields because it is beyond the scope of this paper. To increase sample sizes, we summed several vertical profiles into seasonal average profiles in order to increase the number of observations per altitude. Moreover, to completely remove the model bias from the analysis of the vertical distribution of mercury we calculated a relative vertical profile which we call the mean deviation profile (MDP). The MDP indicates the difference for each individual altitude from the average column concentration and is calculated for models and observations independently. Thus, it indicates whether each model is able to reproduce the observed vertical distribution rather than the actual concentration of mercury species (Eq-3). This is especially valuable for the analysis of oxidized mercury species, as there is an ongoing discussion about an underestimation of concentrations due to limitations of the current measurement techniques (Lyman et al., 2016; Ariya et al., 2015; Gustin et al., 2015; Huang and Gustin, 2015; Jaffe et al., 2014; McClure et al., 2014; Ambrose et al., 2013; Huang et al., 2013; Kos et al., 2013; Lyman et al., 2010) Such a systematical measurement error is canceled out in the calculation of the MDP (Eq. 3).

Individual Layer Mean $\quad \bar{X}_L = \frac{1}{N_L} \sum_{i=1,N_L} X_{(i,L)}$ (Eq. 1)

Total Column Mean $\quad \bar{X} = \frac{1}{M} \sum_{L=1,M} \bar{X}_L$ (Eq. 2)



Mean Deviation Profile $\quad MDP_L = \dfrac{\bar{X}_L - \bar{X}}{\bar{X}}$ (Eq. 3)

$X_{(i,L)}$ model or observation i in layer L

L layer

$N_L$ number of values in layer L

i counter for values in layer L

M number of layers in profile

3. Results and Discussion

Observations indicate that there is a tripartite distribution of total mercury (TM) in
the atmosphere. The highest concentrations (1.4 – 1.8 ng m⁻³) are found inside the
PBL with a strong gradient towards the free troposphere (1.1 – 1.4 ng m⁻³). This
gradient seems to be mainly driven by anthropogenic emissions, as it was not
observed in regions with low primary emissions (e.g. Mace Head, Ireland)
(Sprovieri et al., 2016; Weigelt et al., 2015). Finally, in the stratosphere total
mercury concentrations are typically below 1 ng m⁻³ (0.7 – 1.0 ng m⁻³) (Slemr et
al., 2016; Lyman and Jaffe et al., 2012). The observed TM profiles are often similar
to GEM profiles. Inside the PBL oxidized mercury (RM) (Here, RM is defined as the
sum of all oxidized forms of mercury including GOM and PBM)) concentrations are
very low and mostly between 20 – 100 pg m⁻³ in Europe and North America, even
in source regions (Sonke et al., 2016; Weigelt et al., 2016; 2013; Gay et al., 2013;
Torseth et al., 2012; Prestbo and Gay, 2009). CARIBIC measurements during
intercontinental flights indicate that RM concentrations are also usually below 100
pg m⁻³ in the upper free troposphere (9000 – 12 000m) and only occasionally do
high RM concentrations occur which are probably caused by the direct inflow of
RM from the stratosphere, or the inflow of oxidizing agents which then react with
GEM (Lyman and Jaffe, 2012). A combination of ETMEP and CARIBIC observations
over Germany resulted in a uniform TM and GEM distribution in the free
troposphere during summer (Weigelt et al. 2016) and TM concentrations close to
those measured at ground level were found on 6 overflights of the CARIBIC
aircraft in April, June, and September. A similar vertical distribution was found in

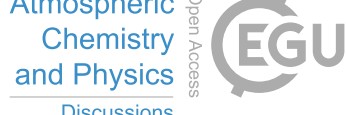



North America during winter (Brooks et al., 2014) and summer (Ambrose et al., 2015; Gratz et al, 2015; Shah et al., 2015). In none of these cases a substantial TM gradient was found inside the free troposphere and the GEM/TM ratio was in the range of 0.95 – 0.99 in the upper free troposphere which is a ratio typically

found inside the PBL. During spring (14th April to 4th June) Brooks et al. (2014) consistently found low TM concentrations above 5000m which indicates a stratospheric intrusion of air masses with low mercury concentrations. Here, the GEM/TM ratio in the upper troposphere decreased to 0.88 to 0.92. For comparison, GEM/TM ratio at the tropopause is around 0.8 – 0.9 and decreases to 0.6-0.8 in the

first 4 km above the tropopause. A similar profile was observed by Gratz et al. (2015) on the 24th June and could be attributed to high bromine concentrations. Bromine as the main oxidizing agent in the upper free troposphere is consistent with findings from CARIBIC that showed no consistent influence of ozone concentrations on the GEM/TGM ratio (Fig. S1).

Finally, in North America a peak of RM concentrations in the range of 100 – 300 pg m$^{-3}$ with GEM/TM ratios below 0.9 was observed in the lower free troposphere (2000 – 4000m). As there are no airborne observations in the range of 3500 – 6500m this feature has not yet been observed over Europe. Possible reasons for the occurrence of this RM peak, which points to GOM production at this altitude,

are still unclear. However, it may be speculated that low relative humidity, low particle surface density, and high solar radiation facilitate photochemistry above the PBL. Based on the findings above, Figure 1 depicts idealized seasonal vertical profiles for the northern mid-latitudes.

Here, we investigate capability of the models to reproduce the observed

atmospheric distribution of TM, GEM, and RM. To increase the sample size for the model evaluation we created seasonal average profiles for Europe and North America. For this, we integrated the high resolution 2.5 minute Tekran data to hourly values, separated all observations into bins of 1000m (0 – 1000, 1000 – 2000, etc.) and calculated the mean concentration as well as the 66% quantile

range for each bin. In addition to the absolute concentrations we investigate mean deviation profiles as described in Section 2.4.



### 3.1 Total and Elemental Mercury

#### 3.1.1 Europe

Based on the combination of ground based observations from the GMOS network (Sprovieri et al., 2016; GMOS, 2016; Weigelt et al., 2013; 2015) and ETMEP observations inside the PBL and the lower troposphere (Weigelt et al., 2016), as well as CARIBIC observations in the upper troposphere and the lower stratosphere

(Slemr et al., 2016) we were able to obtain comprehensive vertical mercury profiles for Europe from the surface up to 12 000m. Here, we present two individual profiles (Figure 2):

The first profile measured on 21$^{st}$ August 11 – 12h UTC at Leipzip, Germany which combines ETMEP and CARIBIC data and was published by Weigelt et al. (2016).

Based on the discussion above and ETMEP GOM measurements being in the range of 20 to 40 pg m$^{-3}$ we expect GEM to be almost identical to TM for these profiles, perhaps except for the data gap in the range of 3000 – 6000m where GOM concentrations could have been higher. It can be seen that the models generally underestimate mercury concentrations. This is in line with many previous model

studies which found that models tend to underestimate current TM concentrations in Europe (Bieser et al., 2014; Chen et al., 2014; Muntean et al., 2014; Gencarelli et al., 2016). However, the majority of model values are still within the measurement uncertainty range (Fig. 2). Looking at the vertical distribution we found that the models are generally able to reproduce the vertical distribution of

both GEM and TGM. However, the PBL height as calculated by the meteorological models has a large influence on the actual altitude of the Hg gradient. It can be seen, for example, that WRF-Chem simulates a PBL height of 500m, while the observations located the top of the PBL at an altitude of 2500m. Here, the PBL growth was delayed in the WRF meteorological model. All models exhibit higher

concentrations inside the PBL and none has a gradient inside the troposphere, which is in agreement with the observations. Concerning the GEM/TGM ratio only one model show values lower than 0.9 – 0.95 inside the troposphere. The





ECHMERIT model exhibits a mostly uniform GEM/TGM ratio between 0.7 and 0.8 over the whole altitude range. This would be a realistic ratio if RM measurements 555 were underestimated by a factor of 5.

Looking at the stratosphere, only the GLEMOS model is able to reproduce a decrease of TM concentrations above the tropopause. Due to the low resolution in this altitude, GLEMOS has only 2 layers between 10 000 and 15 000m, the modeled gradient is less steep than that observed. None of the other models 560 gives significantly lower TM concentrations in the stratosphere. However, GEOS-Chem and GEM-MACH-Hg have increased oxidation above the tropopause. In GEM-MACH-Hg the GEM/TM ratio declines from 0.9 at the tropopause (11 000m) to 0.6 5km above. This is in line with observations from CARIBIC. The GEOS-Chem model also exhibits pronounced mercury oxidation above the tropopause with the 565 GEM/TM ratio declining from 0.9 to 0.1 in the 5km above the tropopause. ECHMERIT and WRF-Chem-Hg have no increased oxidation or reduced TM concentrations above the tropopause. The CMAQ-based models CCLM-CMAQ and CMAQ-Hem have the tropopause as their upper boundary and do not model the stratosphere.

The second profile is a combination of ground based observations at the GMOS station Mace Head, Ireland with the CARIBIC flight of 19[th] September 6 – 7 – 8h UTC (Fig. 2). In 2013, the CARIBIC aircraft passed close to Mace Head six times within a range of 86 – 220km (27[th] April, 28[th] April, 08[th] June, 07[th] June, 19[th] September, 20[th] September) but the other profiles look similar. The CARIBIC data 575 is separated into tropospheric and stratospheric measurements based on the relative height above the tropopause (Sprung and Zahn, 2010). Here, we depict the profile for the nearest CARIBIC overflight. In this region, which is influenced by clean air from the Atlantic Ocean, we did not observe a gradient between the surface and the upper troposphere. Again, models tend to underestimate mercury 580 concentrations. At Mace Head all models are able to reproduce the linear TM concentrations in the free troposphere. However, several models overestimate the concentrations near the surface. It has to be noted, however, that Mace Head is a coastal station with predominantly westerly winds from the open Atlantic which



might be difficult to reproduce for models with a coarse resolution and thus higher
ground based concentrations could be due to anthropogenic emissions from
Ireland. At the tropopause, the observations show an almost instantaneous
decrease of TM concentrations from 1.4 to 1.0 ng m$^{-3}$. The models behave
similarly to the profile over Leipzig with only GLEMOS showing a decrease above
the tropopause. The models with a higher vertical resolution near the tropoause
(GEM-MACH-Hg 12 layer and GEOS-Chem 5 layers between 10 000 and 15 000m)
are better able to reproduce the gradient, albeit they only show a decrease in
GEM/TM ratio not in TM concentration.

As described above we calculated an average summer vertical profile for Europe
using data from 5 ETMEP profiles in Germany and Slovenia performed between
the 19$^{th}$ and 23$^{rd}$ August complemented with CARIBIC flights on the 21$^{st}$ and 22$^{nd}$
August and the 18$^{th}$ and 19$^{th}$ September. Thus, we created an average profile with
290 hourly samples based on a sampling interval of the co-located Tekran
instruments of 2.5 minutes (Fig. 4). We did not use measurements from the Lumex
instrument for this evaluation as none of the other aircraft were equipped with
such an instrument. The performance of the Lumex instrument on this flight is
discussed in Weigelt et al. (2016, this issue). The resulting GEM and TM profiles
are depicted in Figure 3a and 3b respectively. Again, it can be seen that the
models generally underestimate mercury concentrations in central Europe during
August 2013. However, when looking at the mean deviation profile (MDP) which
depicts the relative vertical distribution compared to the total column average
concentration, all the models are within the observed range. Investigating the
experimental model runs, it can be seen that in the case with all anthropogenic
emissions emitted as elemental mercury (ANTSPEC) the models have slightly
higher mercury concentrations near the surface which leads to better agreement
with observed gradients. While all models give similar vertical profiles for the
BASE and ANTSPEC cases, in the cases without anthropogenic emissions (NOANT)
and without atmospheric chemistry (NOCHEM) the models show different
responses. In these cases the modeled vertical distributions of mercury start to
diverge from the observations and each other. This shows the strong impact of





atmospheric chemistry on the vertical GEM distribution and global mercury transport in general.

### 3.1.2 North America

We created similar average vertical mercury profiles for North America based on
185 hourly samples from three profile flights at Tullahoma, TN between 18[th] January 2013 and 14[th] June 2013 (Brooks et al., 2014) (Figure 4) and 898 hourly samples from 7 NOMADSS flights between 20[th] June 2013 and 12[th] July 2013 (Figure 5). For the NOMADSS flights we selected vertical flight paths for this evaluation and discarded horizontal flight paths. Here, the observations exhibit a
similar vertical distribution with higher concentrations inside the PBL and lower concentrations in the FT. The NOMADSS profile contains one flight with a stratospheric intrusion and thus shows a slightly decreasing trend in the upper troposphere. Observed profiles and model results for North America are comparable to Europe. For the summer profile (Fig. 5) there are elevated TM
concentrations inside the PBL and no trend inside the FT. Models tend to underestimate TM and GEM concentrations but are in good agreement with the relative distribution. The higher concentrations near the surface in the ANTSPEC case leads to better agreement with observations. For the winter profile (Fig. 4) GEOS-Chem and GEM-MACH-Hg are in good agreement with the absolute GEM
and TM observations. However, models do overestimate concentrations near the surface, which could be due to modelled PBL height and anthropogenic emission fluxes.

Finally, we created a third profile for spring from three profile flights at Tullahoma, TN on 15[th] April, 10[th] May, and 4[th] June 2013 (Brooks et al., 2014) (Figure 6). This
profile looks different than the others. Again, TM and GEM concentrations are highest inside the PBL but there is a second decreasing gradient between 4000 and 5000m. Above 6000m GEM and TM concentrations fall below 1.0 ng m$^{-3}$ which is a value typically found in the stratosphere. This feature was observed on all three flights during spring and thus seems not to be an individual outlier.
Furthermore, in the time from April to July stratospheric mass transport into the





upper and mid troposphere is known to occur regularly (Appenzeller and Holten, 1996; Allen et al., 2003; Zanis et al., 2003; Olsen et al., 2004; Schoeberl 2004). Moreover, Sprenger et al. (2003) and Sprenger and Wernli (2003) demonstrated that cross tropopause mass flux is highest in the mid latitudes where these

mercury profiles were measured. This is also in line with observations from CARIBIC which found stratospheric intrusions of air masses with low mercury concentrations during this time span (Slemr. p.c.). Stratosphere to troposphere transport of mercury is also the most convincing reason for observed elevated oxidized mercury concentrations in the upper troposphere which is further

discussed in the next Section.

### 3.2 Oxidized mercury

Apart from GEM no individual mercury compound has been identified so far. The speciation of mercury is thus operationally defined as GEM, GOM, and PBM (Gustin

et al., 2015). As the different implementations of the mercury red-ox chemistry in the models presented here is not directly compatible, we decided to sum all oxidized model species for this comparison. Thus, in the following Section we compare modeled reactive mercury RM (RM = GOM + PBM = TM - GEM) concentrations to observations mostly because of the supposed equilibrium

between GOM and GEM (Rutter and Schauer, 2007; Amos et al., 2012). The species measured by the presented aircraft campaigns also differ. Some measure GOM and PBM explicitly and others measure the difference between TM and GEM. Moreover, depending on the sampling inlet geometry and operating conditions, filters in the sampling line, and temperature gradients, a fraction of PBM may not

be accessible to measurement (Slemr et al., 2016). In the following we treat all observations alike and interpret them as total RM measurements.

### 3.2.1 Europe

Measurements at Waldhof, Germany indicate that there is a strong RM gradient

inside the PBL with very low concentrations at the surface and 10 – 15 times higher concentrations above 500m. This is to be expected because of the high





stickiness and therefore fast dry deposition of RM on surfaces (Zhang et al.,
2009). During the ETMEP campaign a total column RM measurement was
performed inside the PBL above the ground based measurement station Waldhof
(Figure 6). Five of the seven models are able to reproduce the RM concentrations
above the surface with one over and one underestimating the concentration. It
has to be noted, that ECHMERIT which strongly overestimates RM is able to
reproduce the low concentrations at the surface and thus is in good agreement
with the relative vertical distribution. An investigation of the experimental model
runs indicated that the overestimation at the surface is due to anthropogenic
emissions and was reduced significantly in the ANTSPEC run while concentrations
above the surface are mainly driven by atmospheric chemistry. This is in line with
the findings of Bieser et al. (2015) and Weigelt et al. (2016).

3.2.2 North America

For North America, we use the same profiles as described in Section 3.1.2. On the
flights at Tullahoma GOM as well as PBM was measured and for the analysis we
plotted the sum as total RM. Due to the long sampling times necessary for
denuder measurements the sample size is much smaller than for the GEM
observations. The winter profiles are based on 32 samples (Fig. 7) and the spring
profiles on 48 samples (Fig. 8).

During winter, RM concentrations varied around 30 pg m$^{-3}$ with slightly lower
concentrations inside the PBL. For the BASE case model results are mostly inside
the uncertainty range of the observations. During winter the models with the
lowest RM production (GEM-MACH-Hg, GLEMOS, CMAQ-Hem) are closest to the
observations. ECHMERIT generally overestimates RM concentrations, while GEOS-
Chem provides increasing concentrations above 4000m which are not in
agreement with observations. This increasing trend was also found in models
when using the GEOS-Chem and p-TOMCAT bromine fields (BRCHEM1 and
BRCHEM2). However, the peak is much more pronounced in the GEOS-Chem run.
Further investigation of the experimental model runs indicates that the amount of
oxidized mercury is strongly dependent on the choice of CTM. For example, the





ECHMERIT model produces the highest RM concentrations for all chemical reactions. With the exception of ECHMERIT all models are closest to the
observations in the BASE case. Looking at the relative vertical distribution, the observations give lower RM concentrations inside the PBL and no trend in the free troposphere. The gradient at the PBL can be reproduced by all chemical reactants but bromine and OH chemistry leads to an increasing trend in the upper troposphere (Fig. 8). Here, only the ozone chemistry is able to reproduce the
observed profiles.

The spring profile for RM at Tullahoma is depicted in Figure 9. Here, a strong RM peak up to 150 pg/m³ can be seen in an altitude of 3000 – 5000m. This peak is above the PBL which was between 2500 and 3200m during these flights which were all made during the afternoon when the PBL reaches its highest expansion.
In the BASE case most models fail to reproduce this peak and only CMAQ-Hem and ECHMERIT, both using ozone chemistry, give similar vertical profiles. On average, the multi-model mean is close to the observed concentrations, but exhibits only the typical gradient at the PBL but no pronounced RM peak. Investigating the relative vertical distribution for different chemistry sensitivity
runs reveals that ozone and OH chemistry are able to reproduce the observed peak. For bromine chemistry the profiles are inverted, exhibiting a minimum where the maximum RM concentrations were observed. Comparing the RM profiles to the TM profiles (Fig. 6) shows that the RM peak is below the presumably stratospheric low TM air masses. This could be an indication that the increased
oxidation is not due to stratospheric bromine transport but due to regional oxidation above the PBL. This would explain, why the bromine chemistry cannot reproduce this peak but ozone and OH chemistry can. Of course it has to be stated that the bromine fields themselves are also subject to large uncertainties and thus the interpretation of these findings depends on the quality of the
bromine fields. However, results are similar for independent bromine datasets from GEOS-Chem and p-TOMCAT bromine fields. Furthermore, there were only two RM measurements which indicate the decline above 6000m and it would also be possible that this peak extended further upwards and was due to a deep





stratospheric intrusion.

Finally, we evaluate the model performance for RM for the summer profile based on NOMADSS data from June and July 2013. Due to the differential measurements approach of the DOHGS instrumental setup the sample size is equal to that of the GEM profiles (Lymann and Jaffe, 2012; Ambrose et al., 2013; Ambrose et al. 2015). The larger sampling size together with the fact that NOMADSS observations cover

a region larger than the vertical profiles over Tullahoma leads to a higher variability in the measurements given by the 66% quantile range (Fig. 10). We created the average RM profile from the same data as the GEM profile. For RM measurements below the detection limit we used half the reported detection limit which varied between 74 and 138 pg m$^{-3}$. Thus, giving us a minimum RM

concentration of 34 pg m$^{-3}$ which is in line with the other observations previously presented.

The resulting profile exhibits a distinct vertical distribution with lower concentrations inside the PBL (40 – 60 pg m$^{-3}$), an RM peak directly above the PBL (100 – 350 pg m$^{-3}$), lower concentrations in the mid-troposphere (50 – 200 pg/m$^3$),

and increasing concentrations in the upper troposphere (100 – 300 pg/m$^3$). The increasing trend in the upper troposphere was attributed to an episode with high bromine concentrations (Gratz ez al., 2015) and accordingly the model runs with bromine chemistry can reproduce this (Fig. 10, BRCHEM). The finding that the ozone and OH reactions cannot reproduce the observed increase in RM

concentrations in the upper troposphere is in line with findings from CARIBIC, where no correlation of ozone with the GEM/TM ratio found (Fig. S1).

Similarly to the spring profile at Tullahoma, the lower RM peak lies directly above the PBL, which is an area of enhanced photolytic activity due to higher solar radiation and low particle density concentrations compared to the PBL. Also, due

to the low water vapor content in this region little aqueous reduction of RM can take place. This RM peak cannot be reproduced by model runs with bromine chemistry. In fact, the resulting profiles are even inverse to the observations. Ozone and OH chemistry on the other hand, lead to increased oxidation above the PBL with the OH chemistry run having the best agreement with the observed





vertical distribution and ozone with the actual concentrations (Fig. 10, O3CHEM,
        OHCHEM).

        3.2.3 Stratosphere

        Stratospheric observations from inter-continental CARIBIC flights indicate that the
GEM/TM ratio declines above the tropopause with values typically in the range
        between 0.6 and 0.8 in the first 4km above the tropopause (Fig. 11). During
        summer values down to 0.5 were found in the tropics. Here, we compare those
        models which include the stratosphere (GLEMOS, GEM-MACH-Hg, GEOS-Chem,
        ECHMERIT) to observations. The models exhibit greater differences in the
stratosphere compared to the troposphere. ECHMERIT exhibits no GEM/TM
        gradient throughout the year with similar values of 0.7 – 0.9 in troposphere and
        stratosphere. Although the model cannot reproduce the declining trend above the
        tropopause, it is mostly within the uncertainty range of the observations.

GLEMOS shows the best agreement with observations. It is able to reproduce the
        slow GEM/TM ratio decrease above the tropopause with values mostly between
        0.5 and 0.7 in the first 4km above the tropopause. GEM-MACH-Hg and GEOS-
        Chem both exhibit much higher oxidation rates in the stratosphere. GEM-MACH-Hg
        also has a slow decrease of GEM/TM ratios above the tropopause but consistently
shows GEM/TM ratios below 0.3 above 12 000m north and south of 30°. Finally, in
        GOES-Chem the GEM/TM ratio decreases earlier, already a few kilometers below
        the tropopause in altitudes of 6000 – 10 000m. Above 12 000m almost all mercury
        is oxidized at the poles and even a the equator the GEM/TM ratio drops below 0.1
        above 16 000m (Fig. 11c). On flights during summer in the range of 30°N – 0°N a
steep decline of the GEM/TM ratio to values below 0.5 was observed, which is in
        line with the profiles modeld by GEOS-Chem. However, it has to be considered
        that the uncertainty of the observations is high and at times no gradient at all was
        observed. The GEM and TM CARIBIC measurements are further discussed in
        Section 3.3.




### 3.3 Inter-hemispheric gradients

Finally, observations on 8 flights from Munich, Germany to Cape Town, South Africa and 19 flights from Munich to Sao Paulo, Brazil are used to investigate the models' capability to reproduce inter-hemispheric gradients. The inter-hemispheric CARIBIC flights were performed between 2013 and 2017. The CARIBIC Tekran instrument, which is usually set up to measure TM, was equipped with a quartz wool filter on each return flight to measure GEM only (Slemr et al., 2016. The Tekran raw data was manually reintegrated (Slemr et al., 2016). This allows us to look at inter-hemispheric gradients of elemental and total mercury. However, as the two quantities were not measured on the same flights only a range of possible oxidized mercury concentrations can be deduced. Long range transport and a variable tropopause height can easily lead to differences larger than the expected RM concentrations on the return flight on the same flight track. Because of this, the calculated average difference of TM and GEM can sometimes be lower than zero. Most of the TM and GEM measurements were within each other's 66% quantile range (Fig. 12a,b). The difference between the average TM and GEM concentrations was 70 pg m$^{-3}$ on the flights to Cape Town (N=756) and 100 pg/m$^3$ on the flights to Sao Paulo (N=1399). A detailed investigation leads to the conclusion that RM concentrations are mostly low (~50pg m$^{-3}$) in the upper troposphere with occasionally high concentrations of up to 200 pg m$^{-3}$ and more. This is in line with the findings presented in Section 3.2, and with three of the four global models which also give an average TM - GEM difference of around 100 pg m$^{-3}$. GLEMOS, GEM-MACH-Hg, and ECHMERIT are in good agreement with observations in the BASE case while GEOS-Chem overestimates oxidized mercury in the mid latitudes (50°N – 30°N), leading to an average of 200 pg m$^{-3}$ (Fig. 12c,d). The results for the sensitivity runs using different chemical reactants leads to similar results and the other models also exhibit increased oxidation in both bromine chemistry runs (Fig. 12g,h).

To create average inter-hemispheric transects we grouped all observations which were at least 1 km below the tropopause into bins of 5° latitude and filtered out high mercury concentrations from polluted air masses (Hg > 2.5 ng/m$^3$). This was





especially necessary on the flights to South Africa where a few large scale biomass burning events lead to measured GEM concentrations of up to 3 ng m⁻³. These events can mask the inter-hemispheric gradient. Finally, the first and last

data points include take-off and landing. This results in a stronger gradient compared to measurements in the upper troposphere.

For the model evaluation we use monthly average GEM and TM concentrations for the month during which each flight was performed from the grid cell closest to the aircraft and aggregate the model data into bins similar to the observational data.

It has to be kept in mind that for models with a low vertical resolution the relevant grid cell might extend above the tropopause. Here, we focus on the relative inter-hemispheric gradient to evaluate the models. The relative TM and GEM trends on flights to Sao Paulo are depicted in Figure 13 and absolute values are given in Figure 14. Similar plots for the fligths to Cape Town are given in the

supplementary material (Figures S2 and S3). The models are generally in better agreement with absolute and relative observations for total mercury (Fig. 13, 14). This is mainly due to an overestimation of oxidized mercury in the northern hemisphere (45°N to 35°N). All models give slightly better results in the ANTSPEC case and the absolute mercury concentrations are 10% higher compared to the

BASE case (Fig. 14c,d). This is consistent with the findings in Section 3.1. In the case without anthropogenic emissions (NOANT) mercury concentrations are much too low and in the NOCHEM run models vastly overestimate mercury concentrations. This is to be expected, as the lifetime of GEM increases without oxidation processes. The exception is the ECHMERIT model which is very close to

observations in the NOCHEM case. This is due to the fact that the ECHMERIT model does not consider dry deposition of GEM. The results in all experimental chemistry runs are strongly dependent on the dynamic response of air-sea exchange. In models that prescribe fixed oceanic emission rates, changing deposition due to changes in the chemistry scheme, cannot be compensated by

re-emissions. The ECHMERIT model for example prescribes fixed oceanic mercury concentrations and thus an increase in deposition will result in lower TM concentrations and vice versa, which explains the very high TM concentrations in



chemistry sensitivity runs. This underlines the importance of the air-sea exchange for global atmospheric models even near the tropopause.

For TM, no chemistry setup could be found that most accurately reproduced the observed concentrations and trends. As was shown before in the evaluation of the vertical profiles, differences in the CTM formulation can have a larger impact than the choice of oxidant. Looking at GEM, it can be see that different oxidants lead to different inter-hemispheric distributions. Here, the use of bromine fields leads to

an overestimation of oxidation in the northern hemisphere (50°N – 25N). On the other hand, the use of ozone and OH chemistry only leads to underestimation of the oxidation around the equator. However, the GEM-MACH-Hg model does not exhibit this feature. With 12 layers between 10 000 and 15 000m the GEM-MACH-Hg model has a much greater vertical resolution around the tropopause compared

to the other models and this has a large impact on model results. In models with coarser vertical resolution, low stratospheric concentrations will have a larger impact on this evaluation. GLEMOS and ECHMERIT are the models with the lowest resolution in this altitude with 2 and 3 layers between 10 000 and 15 000m respectively. GEOS-Chem has 5 layers in this altitude.


## 4. Conclusions

In this model inter-comparison study we investigated the vertical distribution of mercury in the atmosphere and evaluated the impact of mercury chemistry and emissions. The key finding is that models are generally able to reproduce the

vertical profile of total mercury (TM) and elemental gaseous mercury (GEM) from the surface up to the tropopause. This means largely uniform concentrations inside the PBL and free troposphere. Increased GEM concentrations observed inside the PBL could be attributed to anthropogenic emissions. However, the models tend to overestimate GEM concentrations in the lower stratosphere and

those models which feature declining GEM concentrations above the tropopause do so by oxidation to reactive mercury (RM) species, thus overestimating TM. Moreover, it was found that a high vertical resolution near the tropopause is very important for a better reproduction of the observed declining mercury gradient.





The RM, the observations indicate low concentrations inside the PBL, often below 50 pg m$^{-3}$ with a strong decrease towards the surface. This seems plausible due to the high dry deposition velocity of RM. Current model setups tend to overestimate RM near the surface which here could be attributed to the current speciation profiles used for anthropogenic emissions. Also in the FT, most observations are

below 100 pg m$^{-3}$ which is approximately the detection limit of current measurement techniques. Therefore, no further information on possible vertical gradients is available for these regions. However, two separate regions in the upper and lower free troposphere with increased GEM oxidation and RM concentrations above 100 pg m$^{-3}$ up to 500 pg m$^{-3}$ were identified in North

America independently by Brooks et al. (2014) and Ambrose et al. (2013). Because current measurement techniques have been shown to underestimate concentrations of oxidized mercury (Jaffe et al., 2014; Gustin et al., 2015), we have focused the model evaluation on relative vertical distributions which are not subject to systematical measurement errors.

Our interpretation of the observations is that stratospheric intrusions and tropopause folds, which mainly occur during spring time, play an import role for elevated RM concentrations in the upper FT at altitudes above 6000m. The frequency of stratosphere to troposphere transport is regionally variable and has shown to be most common in the latitudes where the measurements were

performed. Supported by the observations of Gratz et al. (2015) the models imply that bromine reactions are responsible for the oxidation of GEM during these episodes. Besides bromine species, stratosphere to troposphere transport could also be a source for RM already formed in the lower stratosphere. This would also explain the missing correlation of ozone concentrations and GEM/TM ratios

measured by the CARIBIC aircraft in the upper FT.

Uniformly low RM concentrations were observed during winter and could be reproduced by the models. In spring and summer, increased RM concentrations were observed above the PBL in the lower free troposphere were observed during

spring and summer. This could only be reproduced by models using $O_3$ and OH chemistry. Any oxidant directly above the PBL is either produced locally or transported from the PBL and thus OH seems a plausible explanation. Moreover, reduced water vapor content and particle surface densities would reduce any occurring aqueous RM reduction processes.


Finally, we have investigated TM and GEM concentrations and gradients in the upper troposphere between the northern and southern hemisphere based on inter-continental CARIBIC flights. The models were more adept in reproducing TM concentrations and trends compared to GEM. Model runs using bromine reactions
showed a better agreement to observed inter-continental TM gradients. However, the current bromine fields lead to a strong overestimation of mercury oxidation in mid-latitudes. Ozone and OH chemistry, on the other hand, led to overestimated oxidation in the tropics. Interestingly, reducing the RM fraction in the anthropogenic emission inventories led to a better agreement with observed
concentrations. This could be due high RM fractions for coal fired power plants in current emission inventories which have high stacks and thus effective emission heights can even be above the PBL at times.

   List of contributors:
Modelling:
   GLEMOS: Oleg Travnikov
   GEOS-CHEM: Noelle Selin, Shaojie Song
   GEM-MACH: Ashu Dastoor, Andrei Ryskov
   ECHMERIT: Francesco De Simone, Ian Hedgecock
CMAQ-Hem: Che-Jen Lin, Yun Zhu
   WRF-CHEM: Christian Gencarelli, Ian Hedgecock
   CCLM-CMAQ: Johannes Bieser, Volker Matthias, Beate Geyer
   p-TOMCAT: Xin Yang

Observations:
   ETMEP: Andreas Weigelt, Johannes Bieser, Ralf Ebinghaus, Nicola Pirrone
   NOMADSS: Daniel Jaffe, Jesse Ambrose, Lynne Gratz





Tullahoma: Steve Brooks, Xinrong Ren, Winston Luke, Paul Kelley

CARIBIC: Carl Brenninkmeijer, Andreas Zahn, Franz Slemr, Andreas Weigelt, Ralf Ebinghaus

Waldhof: Andreas Weigelt, Ralf Ebinghaus

Mace Head: Andreas Weigelt, Ralf Ebinghaus, Nicola Pirrone

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



|  | GLEMOS | GEOS-CHEM | GEM-MACH-Hg | ECHMERIT | CMAQ-Hem | WRF-CHEM | CCLM-CMAQ |
|---|---|---|---|---|---|---|---|
| **Spatial resolution** Scope Horizontal Vertical | Global 1° x 1° 20 levels, top 10 hPa | Global 2.5° x 2° 47 levels, top 0.01 hPa | Global 1° x 1° 58 levels, top 7 hPa | Global T42 (~ 2.8° x 2.8°) 19 levels, top 10hPa | Hemispheric 108 x 108 km² 13 levels, top 50 hPa | Regional 24 x 24 km² 30 levels, top 50 hPa | Regional 24 x 24 km² 30 levels, top 50 hPa |
| **Meteorology** Data support type Meteorological driver | off-line WRF 3.7.2 / ECMWF | off-line GEOS-FP | on-line GEM | on-line ECHAM5 | off-line WRF 3.7 / NCEP | on-line WRF 3.4 / NCEP | off-line CCLM 4.8 / NCEP |
| **Anthropogenic emissions** Emission inventory Average speciation GEM : GOM : PBM | AMAP 81 : 15 : 4 | AMAP 71 : 19 : 0 | AMAP 96 : 3 : 1 | AMAP 81 : 15 : 4 | AMAP 87 : 10 : 3 | AMAP | AMAP 94 : 1 : 5 |
| **Natural emissions** |  |  |  |  | - |  | Bash et al., |
| **Boundary conditions** mercury other species | - - | - - | - - | - | GEOS-Chem GEOS-Chem | ECHMERIT MOZART-4 | GLEMOS TM-5 |
| **BASE chemistry** gas phase aqueous phase reduction processes | Ozone, OH HOCL/OCL yes | Bromine OH no | OH, Bromine[a] - no | Ozone, OH Ozone, OH no | Ozone, OH Ozone, OH yes | Ozone, OH Ozone, OH Ozone, OH no | Ozone, OH Ozone, OH yes |
| **References** | Travnikov and Ilyin | Holmes et al. (2010); | Durnford et al (2012); | Jung et al. (2009); De | Byun and Chang | Grell et al. (2005); | Byun and Chang |





| | | | | | | |
|---|---|---|---|---|---|---|
| (2009); Travnikov et al. (2009) | Amos et al. (2012); Song et al. (2015) | Kos et al. (2013); Dastoor et al. (2015) | Simone et al. (2014) | (1999); Byun and Schere (2006); Bullock and Brehme (2002); Bullock et al. (2006); Pongprueksa et al. (2011) | Gencarelli et al., (2014; 2015) | (1999); Byun and Schere (2006); Bullock and Brehme (2002); Bullock et al. (2006); Bash et al. (2010); Bieser et al. (2015) |

Table 1: Model description

| Name | Anthropogenic emissions | Gas-phase chemistry | Description |
|---|---|---|---|
| BASE | UNEP2010 | Model standard configuration | Base run |
| NOANT | No emissions | Model standard configuration | Effect of antrhopogenic emissions |
| ANTSPEC | UNEP2010, 100% GEM | Model standard configuration | Effect of emission speciation |
| NOCHEM | UNEP2010 | No chemistry | Effect of chemistry |
| OHCHEM | UNEP2010 | GEM oxidation by OH | OH dataset from MOZART |
| O3CHEM | UNEP2010 | GEM oxidation by O3 | O3 dataset from MOZART |
| BRCHEM1 | UNEP2010 | GEM oxidation by Br | Br dataset from GEOS-Chem |
| BRCHEM2 | UNEP2010 | GEM oxidation by Br | Br dataset from p-TOMCAT |

Table 2: Specification of model experiments






*Figure 1: Idealized observed TM and GEM mercury profiles for winter, spring, and summer in northern mid-latitudes.*


*Figure 2: Upper panel: GEM/TGM profiles at Leipzig, Germany (21$^{st}$ August 2013) compiled from ETMEP and CARIBIC measurements (Weigelt et al., 2016). Lower panel: GEM/TGM profiles at Mace Head, Ireland (19$^{th}$ September 2013) compiled from GMOS ground based observations (Weigelt et al., 2015) and CARIBIC measurements (Slemr et al., 2016). Solid lines indicate total mercury (TM), dashed lines indicate elemental mercury (GEM), and dotted lines depict the GEM/TM ratio given on the second x-axis.. The horizontal gray lines depict PBL and tropopause height. The black squares are ETMEP measurements, the gray circles are*

*tropospheric and the gray squares are stratospheric CARIBIC measurements.*

*Figure 3 Comparison of modelled average mercury profile for Europe to observations based on vertical profiles from ETMEP and CARIBIC campaigns amended with ground based observations at Waldhof and Mace Head (Weigelt et al., 2013; Slemr et al., 2016). The error bars indicate the 66% quantile range of the observations in*

*each altitude, the sample size for each altitude is indicated on the y-axis of the legend.*

*Figure 4: Comparison of modelled average mercury profile for North America to observations based on vertical profiles at Tullahoma, TN from January and February 2013 (Brooks et al., 2014). The error bars indicate the 66% quantile range of the observations in each altitude, the sample size for each altitude is indicated on the y-*

*axis of the legend.*

*Figure 5: Comparison of modelled average mercury profiles for North America to observations based on NOMADSS flights in June and July 2013 (Shah et al., 2015; Gratz et al., 2016). The error bars indicate the 66% quantile range of the observations in each altitude, the sample size for each altitude is indicated on the y-axis of*

*the legend.*

*Figure 6: Comparison of modelled average mercury profile for North America to observations based on vertical profiles at Tullahoma, TN from April to June 2013 (Brooks et al., 2014). The error bars indicate the 66% quantile range of the observations in each altitude, the sample size for each altitude is indicated on the y-axis of*

*the legend.*

*Figure 7: GOM profiles at Waldhof Germany (23$^{rd}$ August 2013) (Weigelt et al., 2016). The observations are a combination of ground based measurements and a total column measurement in altitudes from 500m to 3000m. Model values are given for BASE (solid line), ANTSPEC (dashed line), NOCHEM (dotted line).*






*Fig. 8: Comparison of modelled average reactive mercury profiles (RM = GOM + PBM) with observations at Tullahoma, TN for January and February 2013 reported by Brooks et al. (2014). The errorbars indicate the 66% quantile range of the observations in each altitude, the sample size for each altitude is indicated on the y-axis of*

*the legend.*

*Fig. 9: Comparison of average reactive mercury profiles (RM = GOM + PBM) at Tullahoma, TN for April, May, and June (Brooks et al., 2014). The errorbars indicate the 66% quantile range of the observations in each altitude, the sample size for each altitude is indicated on the y-axis of the legend.*


*Figure 10: Comparison of modelled average reactive mercury (RM = GOM) concentration to observations based on NOMADSS flights in June and July 2013 (Shah et al., 2015; Gratz et al., 2016). The errorbars indicate the 66% quantile range of the observations in each altitude, the sample size for each altitude is indicated on the y-axis of the legend.*


*Figure 11: Seasonal vertical profiles of modeled GEM/TM ratios for winter (upper panel) and summer (lower panel). Observations are based on TM and GEM measurements from CARIBIC flights.*

*Figure 12: Average inter-hemispheric transects for 19 flights from Munich to Sao Paulo. TGM was measured on*
*the outward and GEM on return flights (Slemr et al., 2014). Error bars indicate the 66% quantile range of all observations for a given latitude. Plots in the left column are for TGM and those on theright for GEM.*

*Figure 13: Relative inter-hemispheric transects for 19 flights from Munich to Sao Paulo. TM (left side) was measured on the outward and GEM (right side) on return flights (Slemr et al., 2014). Error bars indicate the*
*66% quantile range of all observations for a given latitude. Plot in the left column are for TGM and in the right side for GEM.*

*Figure 14: Average inter-hemispheric transects for 19 flights from Munich to Sao Paulo. TM (left side) was measured on the outward and GEM (right side) on return flights (Slemr et al., 2014). Error bars indicate the*
*66% quantile range of all observations for a given latitude.*





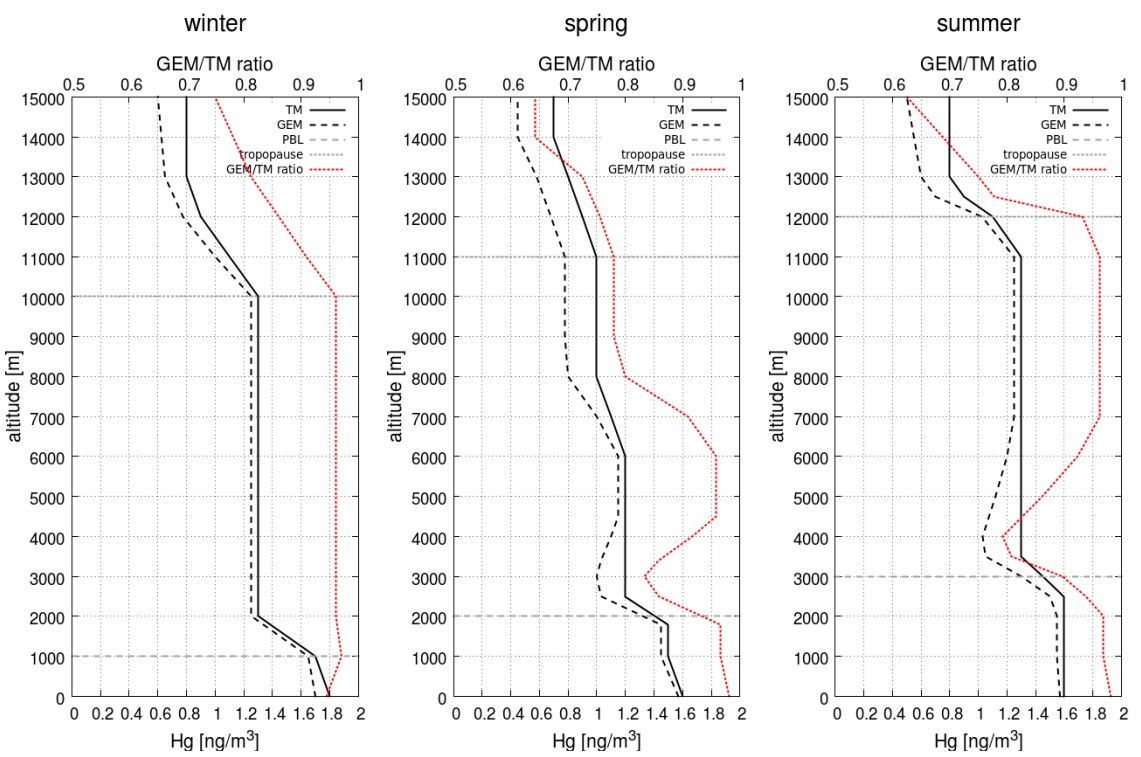

*Fig. 1*





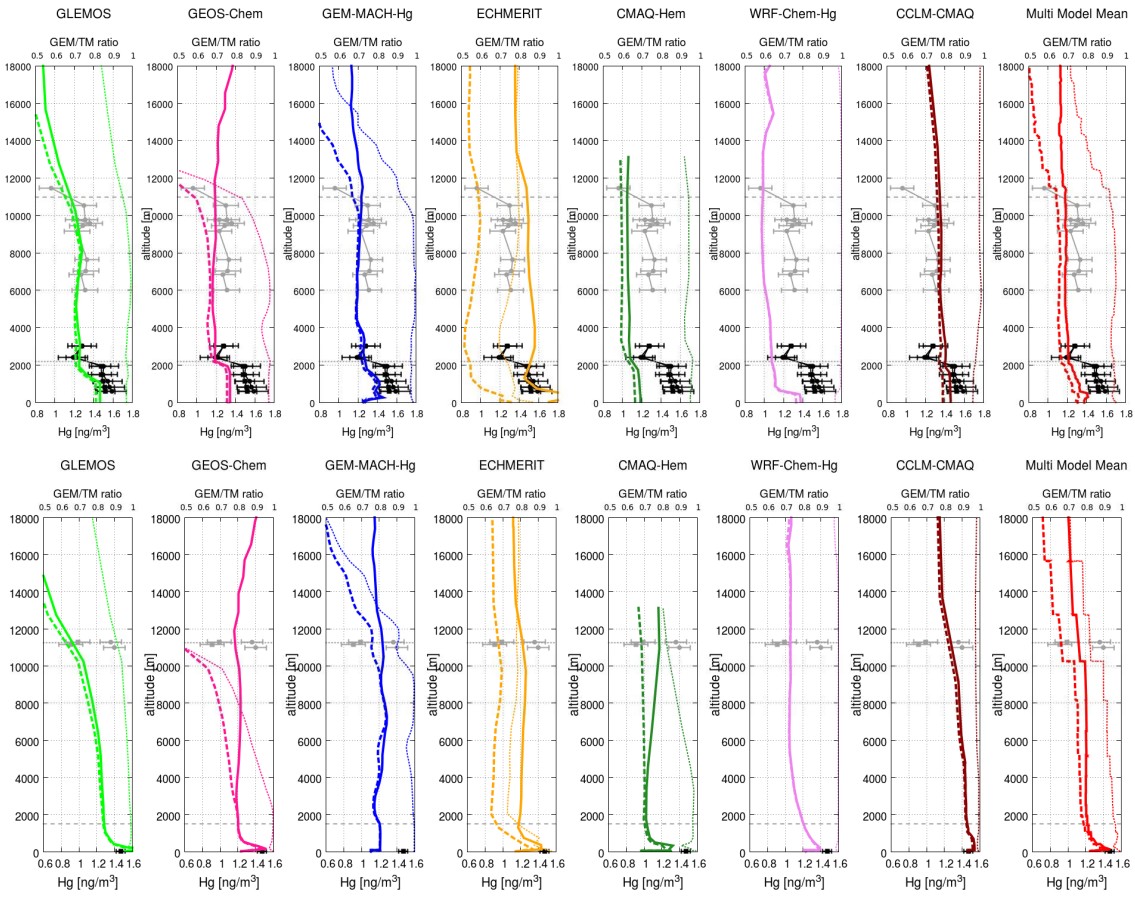

*Fig. 2*





*Fig. 3*

*Fig. 4*





*Fig. 5*

*Fig. 6*





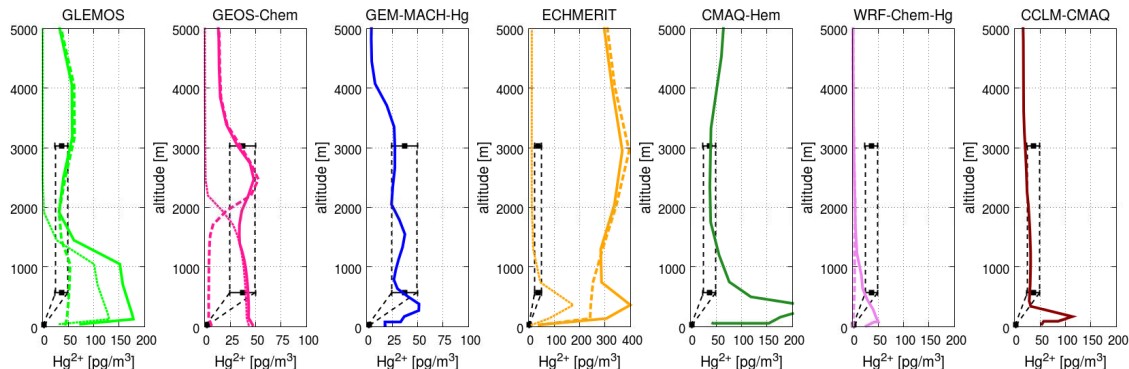

*Fig. 7*





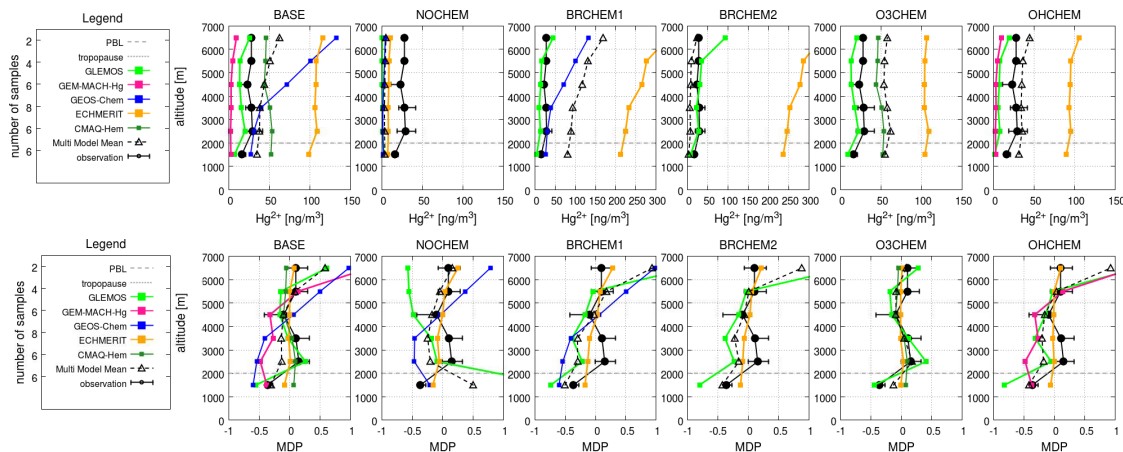

*Fig. 8*

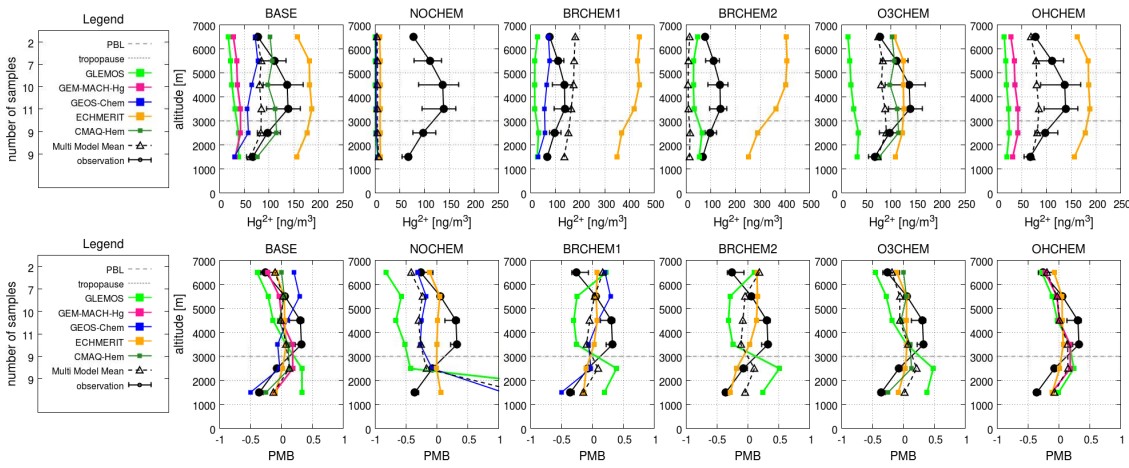

*Fig. 9*

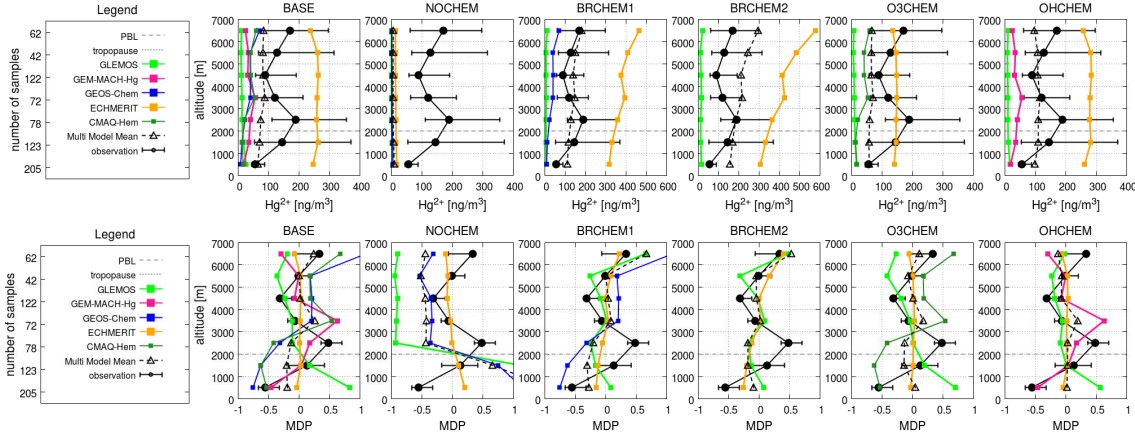

*Fig. 10*





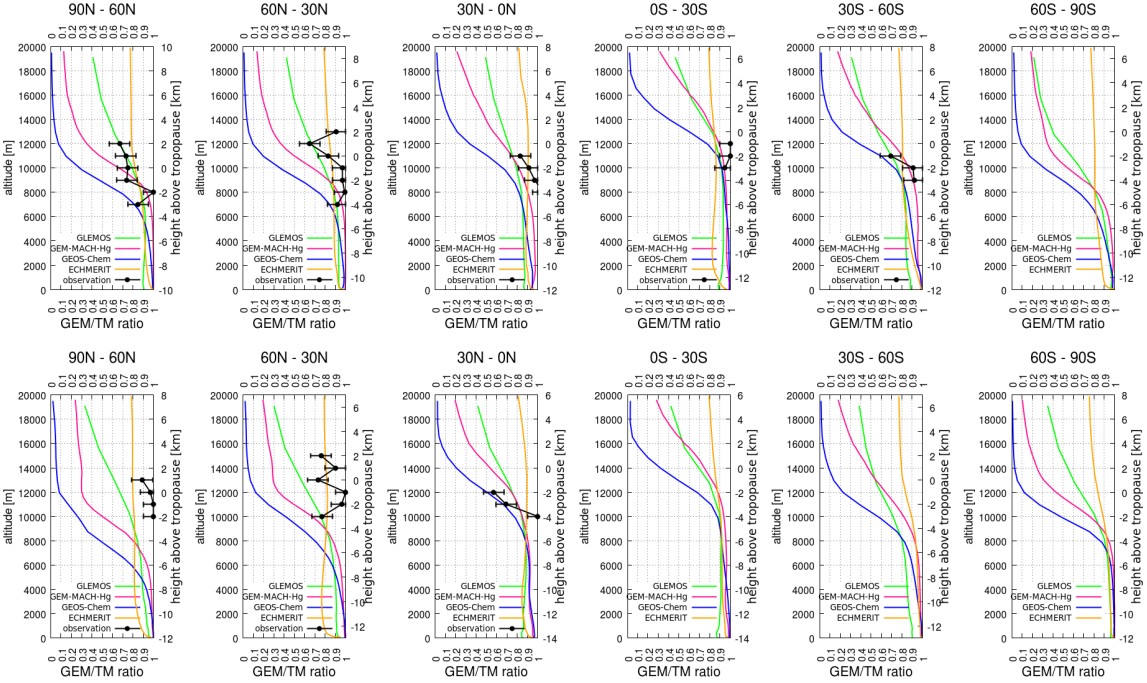

*Fig. 11*



*Fig. 12*



*Fig. 13*





*Fig. 14*