# Peer review of "Multi-model study of mercury dispersion in the atmosphere: Vertical and inter-hemispheric distribution of mercury species"

_Atmospheric Chemistry and Physics, 2016_

## Referee Comment (RC1) · Anonymous Referee #1 · 29 Dec 2016

The manuscript examines the vertical profiles and inter-hemispheric gradients of mercury species observed during three recent aircraft campaigns using an ensemble of seven atmospheric mercury models. The study also evaluates the response of the models to the simulated chemistry and emission speciation, two processes that are poorly understood. This study beings together a large set of models and observations to understand these processes and is an important contribution to the field. My comments on the manuscript are as follows:

General comments: 1) A large portion of the manuscript is focused on comparing model results to the observations, but the authors do not provide any numerical measure of agreement between the model and observations. There are several qualitative

comparisons, but the lack of numerical comparison makes it very difficult for the readers to draw their own conclusions. I recommend the authors include one or more of the standard metrics (mean bias, mean error, correlation coefficient, etc.) for model-observation comparisons.

2) The manuscript lacks a discussion of the representativeness of the observations. I understand that these are the best observations we have, but, as the authors also point out on Line 116-117, aircraft based observations are not representative. Yet they seem to ignore the limited temporal and spatial coverage of the observations when they construct vertical profiles of TM and GEM for the northern mid-latitudes (Fig. 1). It is not clear how these profiles were calculated. Secondly, it is not clear for what time period was simulated by the models, how were the models sampled, and what steps were taken to address the issue of representativeness when making comparisons between the model and the observations.

3) It seems to me that the manuscript was not thoroughly proof-read before publication in ACPD. There are several minor errors that often are distracting. For example, the citation 'Lyman and Jaffe, 2012', was cited at times as 'Lyman and Jaffe, 2011', 'Lyman and Jaffe et al., 2012', 'Lymann and Jaffe, 2012', and 'Jaffe and Lyman, 2012'). There are also a few instances where abbreviations were used without prior definitions. For example, in the main text RM was used first on Line 515, but not defined until Line 663, while in the abstract it was referred to as oxidized mercury, but in the main text as reactive mercury. I recommend that the manuscript be thoroughly proof-read before final submission.

Specific comments:

1) Sect. 2.3: What was the spin-up period for the sensitivity simulations?

2) Lines 452-457: The underestimte in GOM concentrations seems to be related to the ambient absolute humidity and ozone concentrations and is likely not systematic, as stated by the authors.

3) Line 543: Do the authors mean measured 'variability' instead of 'uncertainty'? Same for Lines 699, 797.

4) Line 567: It is stated that CCLM-CMAQ has the tropopause as its upper boundary, but Fig. 2 shows model values for CCLM-CMAQ upto 18 km.

5) Line 580: 'Linear TM'. This term is not clear.

6) Lines 910-920: The authors interpret the high RM above 6 km as being related to stratospheric transport of Br and cite Gratz et al. (2015). However, Gratz et al. (2015) did not find evidence of stratospheric intrusion, and the authors' conclusions seem contradictory to that study. Is it possible to reconcile these two interpretations?

7) Line 927: 'OH seems a plausible explanation'. How about O3? The models with O3 chemistry had better agreement with the observed concentrations for the NOMADSS profile.

8) Lines 940-942: I am not sure how the higher effective height of emissions would affect GEM concentrations in the upper troposphere.

9) Table 1: Is OH an aqueous phase oxidant of GEM in the GEOS-Chem model?

10) Table 2: Since the results of the sensitivity simulations were not available for all models, I suggest adding a column specifying which models participated in each sensitivity run.

11) Title: Perhaps the title should contain "Vertical and inter-hemispheric distribution of mercury species".

Technical comments:

1) Lines 83-84: Grammar

2) Lines 98-102: Grammar

3) Line 143: Spelling, 'Woll'

4) Line 144: What does DOHGS stand for?

5) Line 620: '14th June' seems to be typo.

6) Line 665: Do the authors mean equilibrium between GOM and PBM?

7) Table 1: No emission speciation information for WRF-Chem

8) Table 1: No reference for natural emissions for the first six models.

9) Table 1: Missing footnote 'a'.

10) Table 2: Refers to the emission set as UNEP2010, while Table 1 refers to it as AMAP.

11) Fig. 3: Lower panel. Is MDP for TM or GEM?

12) Fig.7-9: Is Hg2+ different from RM?

13) Figs. 8-9: Units for Hg2+.

14) Fig. 8: Should the x-axis label be MDP instead of PMB?

---

## Referee Comment (RC2) · Anonymous Referee #2 · 4 Jan 2017

The vertical distribution of mercury (Hg) in the atmosphere is an important aspect of studies on global atmospheric Hg cycling. Many previous studies have observed clear vertical gradients of Hg species in the global atmosphere, which were thought to be influenced by atmospheric physicochemical transformation and atmospheric transport processes. This paper conducted a comprehensive modeling study and compared it with field observations on a global scale. This paper provides a significant contribution to current research questions. The modeling results are generally in line with recent observations from air-craft campaigns and at high-altitude sampling sites, indicating conversions of atmospheric Hg species is a global phenomenon. The paper is well written and the methods and discussions are overall credible. I suggest to publish the

paper after minor revisions and clarifications: 1) Please clarify whether the heights mentioned throughout the manuscript are referred to the height above sea level (asl) or above ground level (agl). If the height refers to als, the authors should also compare their modeling results with observations at high-altitude peaks worldwide (e.g., Mount Bachelor Observatory, USA, 2700 m asl; Storm Peak, USA< 3200 m asl; Lulin Atmospheric Background Station, Taiwan, 2862 m asl; Pic du Midi Observatory, France, 2877 m asl and Mt. Leigong, China, 2178 m asl. These observations are in the free troposphere and can be compared with the modeling results). 2) Line 480: Please clarify the mean of 'source regions'. Are they related to anthropogenic or natural sources (GOM and PBM formation in the atmosphere)? 3) Line 480: the citation should be Fu et al., 2016. 4) The authors modeled the vertical concentrations of GEM, GOM, and PBM in the troposphere and stratosphere. Will it be possible to give the total quantity (Mg) of GEM, GOM, and PBM in the PBL, lower free troposphere, middle free troposphere, upper free troposphere and stratosphere? 5) The atmospheric physicochemical properties over the oceans and continents are generally different. I suggest the authors should also calculate the average vertical distributions of atmospheric Hg species over oceans and continents. 6) The measurements of GOM and PBM have many uncertainties. As mentioned the in the paper, previous studies for GOM and PBM measured utilized several different techniques. The authors should introduce the uncertainties of these observations and the effects on their comparisons. 7) Line 1418, Shah et al., 2015 should be Shah et al., 2016. 8) GEM, GOM, and PBM (generally the Hg bounded with fine particulates) are the three major forms of atmospheric Hg. In many parts of the paper, the authors used TM (total atmospheric Hg) and RM (reactive atmospheric Hg), and this is not completely right under some situations. Hg bounded with coarse particulates could represent a large fraction of total particulate Hg in the PBL. Also, GEM, GOM, and PBM could be transformed to other Hg species including Hg in cloud vapor, fog, etc.. These Hg species in the atmosphere sometimes represents an important fraction of atmospheric Hg. Should we define these species as RM? Have the authors taken these species into the modeling? I think this might be an important

element influencing the comparisons between observations and modeling.

---

## Referee Comment (RC3) · Anonymous Referee #3 · 13 Jan 2017

**Summary:** The study by Bieser et el. reports multi-model study of mercury dispersion in the atmosphere: Vertical distribution of mercury species. Honestly speaking, this is a very important topic in global mercury studies as well as this study shares light on assessing the global/regional mercury transport or mass balance study by different models. In addition, this study is well structured and also in a good English writing. I recommend that this manuscript be published in ACP GMOS Special Issue after the authors address these comment.

**General comment**

Only the CCLM-CMAQ model considered the natural emission inventory. I would like to ask the authors to discuss the influence without the natural emission for these model simulation when compared to CCLM-CMAQ, and observed vertical atmospheric Hg profiles in more details.

**specific comments**

Line 513 " Figure 1 depicts idealized seasonal vertical profiles for the northern mid-latitudes." Please specify the sources.

Line 539-541 "This is in line with many previous model studies which found that models tend to underestimate current TM concentrations in Europe" , can be caused by the inventory or modeling setup ? Please give more detailed discussion for this.

Line 657-660 "Apart from GEM no individual mercury compound has been identified so far. The speciation of mercury is thus operationally defined as GEM, GOM, and PBM (Gustin et al., 2015)." In my opinion, this sentence should be removed into introduction section.

Line 680 "Five of the seven models", please specify these five models.

Line 690-695, please discussed the uncertainties of the GOM and PBM measured by Tekran and site the paper from Gustine's group recently before comparing the observation and simulated results.

---

## Author Comment (AC1) · 6 Apr 2017

Answers to Reviewer #3:

We want to thank reviewer #3 for his comments. We implemented all suggestions into the revised manuscript.

Q: Only the CCLM-CMAQ model considered the natural emission inventory. I would like to ask the authors to discuss the influence without the natural emission for these model simulation when compared to CCLM-CMAQ, and observed vertical atmospheric Hg profiles in more details.

A: All models used natural emissions of Hg. However, there was an error in Table 1.

We corrected this and Table 1 now includes the natural emission totals as used by the all models.

Q: Line 513 " Figure 1 depicts idealized seasonal vertical profiles for the northern mid-latitudes." Please specify the sources.

A: The depicted profiles are based on aircraft observations from CARIBIC, ETMEP, NOMADDS, and Tullahoma flights. Data gaps in altitudes in which no observations are available were estimated (we added this information to the figure caption)

Q: Line 539-541 "This is in line with many previous model studies which found that models tend to underestimate current TM concentrations in Europe" , can be caused by the inventory or modeling setup ? Please give more detailed discussion for this.

A: We now discuss this in more detail. "Based on a model run from 1996 to 2008 Muntean et al. (2014) hypothesized that this was due to an overestimation of emission reductions in the last decade. Moreover, a change in the speciation of mercury emissions due to new cleaning technologies of modern coal fired power plants can have an impact on the lifetime of regional primary anthropogenic emissions."

Q: Line 657-660 "Apart from GEM no individual mercury compound has been identified so far. The speciation of mercury is thus operationally defined as GEM, GOM, and PBM (Gustin et al., 2015)." In my opinion, this sentence should be removed into introduction section.

A: We moved this to the introduction: "However, apart from GEM no individual mercury compound has been identified so far and the atmospheric oxidized mercury is an unknown mixture of mercury bound to Br, Cl, OH, O, and NO2 compounds (Horowitz et al, 2017). The speciation of mercury is thus operationally defined as GEM, GOM, and PBM (Gustin et al., 2015). In the following we will address the sum of all oxidized mercury species as OM (oxidized mercury)."

Hannah M. Horowitz, H.M., Jacob, D.J., Zhang, Y., Dibble, T.S., Slemr, F., Amos, H.M.,

Schmidt, J.A., Corbitt, E.S., Marais, E.A., Sunderland, E.M., 2017. A new mechanism for atmospheric mercury redox chemistry: Implications for the global mercury budget. Atmos. Chem. Phys. Discuss., doi:10.5194/acp-2016- 1165, 2017

Q: Line 680 "Five of the seven models", please specify these five models.

A: We now name the models: (GLEMOS, GEOS-Chem, GEM-MACH-Hg, CMAQ-Hem, CCLM-CMAQ)

Q: Line 690-695, please discussed the uncertainties of the GOM and PBM measured by Tekran and site the paper from Gustine's group recently before comparing the observation and simulated results.

A: We added a pragraph at the end of section 3.2: "As discussed in Section 2.4, current GOM measurement techniques which are based on the sorption of GOM on KCl coated denuders have been shown to be susceptible to environmental interferences. Mainly, ozone and humidity have shown to lead to an underestimation of ambient GOM concentrations (Lyman et al., 2010; Jaffe et al., 2014; Gustin et al., 2015). Thus, we focus the following model evaluation on the relative distribution of OM in the atmosphere rather than absolute values."
* * *

---

## Author Comment (AC2) · 11 Apr 2017

Answers to Reviewer #1:

We want to thank reviewer #1 for the valuable feedback. We improved the manuscript based on these suggestions:

1) A large portion of the manuscript is focused on comparing model results to the observations, but the authors do not provide any numerical measure of agreement between the model and observations. There are several qualitative comparisons, but the lack of numerical comparison makes it very difficult for the readers to draw their own conclusions. I recommend the authors include one or more of the standard metrics (mean bias, mean error, correlation coefficient, etc.) for model-observation comparisons.

A: We agree with the reviewer and added a quantitative comparison to the manuscript. For GEM, we chose to use the mean normalized bias (MNB) and mean normalized error (MNE). We prefer these to the normalized mean bias as it gives more weight to the individual data points. We added three tables to the manuscript: Table 1 gives MNB and MNE for each model separated Europe and North America. Table 2 gives model ensemble MNB and MNE values for altitude slices of 1000m in order to identify whether the models perform better or worse in different altitudes. We found that the model bias and error are mostly uniformly distributed in the troposphere with larger errors in the lower stratosphere. We did find a bias minimum between 2000 and 3000m which we think is an artifact due to the observed decrease of GEM concentrations above the PBL which was mostly in this altitude range. Moreover, for GEM bias and error do not differ significantly between Europe and North America.

| Region | Europe | | North America | |
|---|---|---|---|---|
| Species | GEM | | GEM | |
| Model | MNB | MNE | MNB | MNE |
| GLEMOS | -0.07 | 0.16 | -0.12 | 0.16 |
| GEOS-Chem | -0.18 | 0.21 | -0.11 | 0.16 |
| GEM-MACH-Hg | -0.04 | 0.15 | 0.08 | 0.17 |
| ECHMERIT | -0.27 | 0.34 | -0.27 | 0.28 |
| CMAQ-Hem | -0.20 | 0.27 | -0.23 | 0.25 |
| WRF-Chem | -0.17 | 0.25 | - | - |
| CCLM-CMAQ | 0.05 | 0.19 | - | - |
| ENSEMBLE | -0.14 | 0.21 | -0.13 | 0.20 |

Table 1: Mean normalized bias and mean normalized error for each model as well as the model ensemble For GEM in Europe and GEM and OM in North America.

| altitude | Europe | | North America | |
|---|---|---|---|---|
| | MNB | MNE | MNB | MNE |
| 0 - 1000m | -0.20 | 0.20 | -0.17 | 0.19 |
| 1000 – 2000m | -0.22 | 0.23 | -0.21 | 0.25 |
| 2000 – 3000m | -0.08 | 0.15 | -0.12 | 0.17 |
| 3000 – 4000m | -0.14 | 0.16 | -0.16 | 0.20 |
| 4000 – 5000m | -0.21 | 0.21 | -0.11 | 0.21 |
| 5000 – 6000m | -0.27 | 0.27 | -0.04 | 0.24 |
| 6000 – 7000m | -0.20 | 0.24 | -0.12 | 0.24 |
| 7000 – 8000m | -0.28 | 0.28 | - | - |
| 8000 – 9000m | -0.28 | 0.28 | - | - |
| 9000 – 10000m | -0.24 | 0.24 | - | - |
| 10000 - 11000m | -0.26 | 0.26 | - | - |
| 11000 - 12000m | -0.24 | 0.25 | - | - |
| > 12000m | 0.33 | 0.41 | - | - |

Table 2: Model ensemble vertical distribution of model mean normalized bias and mean normalized error for GEM in Europe and North America.

Finally, we calculated the correlation for the vertical oxidized mercury profiles. The results underline the findings already discussed with better performance of Br chemistry for the NOMADSS campaign and better performance of OH and O3 chemistry for the Tullahoma flights.

| | Tullahoma flights January and February (Fig. 8) | | | | | |
|---|---|---|---|---|---|---|
| | BASE | NOCHEM | BRCHEM1 | BRCHEM2 | O3CHEM | OHCHEM |
| GLEMOS | 0.76 | -0.84 | 0.46 | 0.47 | 0.82 | 0.56 |
| GEOS-Chem | 0.37 | 0.16 | 0.37 | | | |
| GEM-MACH-Hg | 0.23 | | | | | 0.23 |
| ECHMERIT | 0.77 | 0.49 | 0.40 | 0.40 | 0.42 | 0.55 |
| CMAQ-Hem | -0.10 | | | | -0.10 | |
| | Tullahoma flights April, May, June (Fig. 9) | | | | | |
| | BASE | NOCHEM | BRCHEM1 | BRCHEM2 | O3CHEM | OHCHEM |
| GLEMOS | -0.17 | -0.59 | -0.80 | -0.71 | -0.21 | 0.37 |
| GEOS-Chem | 0.39 | -0.62 | 0.39 | | | |
| GEM-MACH-Hg | 0.63 | | | | | 0.63 |
| ECHMERIT | 0.93 | 0.17 | 0.54 | 0.52 | 0.87 | 0.94 |
| CMAQ-Hem | 0.53 | | | | 0.53 | |

| | NOMADSS flights (Fig. 10) | | | | | |
| --- | --- | --- | --- | --- | --- | --- |
| | BASE | NOCHEM | BRCHEM1 | BRCHEM2 | O3CHEM | OHCHEM |
| GLEMOS | -0.55 | -0.60 | 0.08 | 0.03 | -0.49 | -0.54 |
| GEOS-Chem | 0.35 | -0.49 | 0.35 | | | |
| GEM-MACH-Hg | 0.07 | | | | | 0.07 |
| ECHMERIT | -0.05 | -0.44 | 0.43 | 0.39 | -0.05 | 0.03 |
| CMAQ-Hem | 0.13 | | | | | 0.13 |

Table 3: Correlation of individual models for OM profiles depicted in Figures 8, 9, and 10.

2.1) The manuscript lacks a discussion of the representativeness of the observations. I understand that these are the best observations we have, but, as the authors also point out on Line 116-117, aircraft based observations are not representative. Yet they seem to ignore the limited temporal and spatial coverage of the observations when they construct vertical profiles of TM and GEM for the northern mid-latitudes (Fig. 1). It is not clear how these profiles were calculated.

A: So far only single profiles for the vertical Hg distribution were published, mostly only considering GEM or TGM. As the presented manuscript is the first comprehensive global analysis of the vertical distribution of mercury we decided to combine all available aircraft based observations to estimate idealized vertical profiles. These profiles represent our current best knowledge of the distribution of mercury species in the atmosphere and we think that they are an important contribution to the scientific discourse.

We added a paragraph discussing this at the end of Section 2.1:

*"These flights cover a large horizontal area, namely the mid latitudes in Europe (45°N - 55°N) and North America (30°N – 45°N) and a large vertical area ranging from the surface up to the lower stratosphere (12000 m). Moreover, comparable flights were performed throughout the year between January and October. Finally, all measurements were performed with Tekran instruments allowing for a comparison of all aircraft based measurements as well as the combination with ground based observations which use similar instruments. It is arguable whether this is already enough data to give us a comprehensive and representative picture of the vertical distribution of mercury in the atmosphere. However, we think that there is an adequate amount of data to allow for more than just an anecdotal investigation of a specific episode. Thus, we combined measurements from all flights in Europe and North America as well as ground based observations for the year 2013 in order to construct idealized seasonal average vertical profiles for TM and OM (Fig. 1)…."*

2.2) Secondly, it is not clear for what time period was simulated by the models, how were the models sampled, and what steps were taken to address the issue of representativeness when making comparisons between the model and the observations.

A: We added more information to the model evaluation section 2.4:
"For the model evaluation we used hourly model results for the year 2013 for all models, with the exception of ECHMERIT which provided a lower temporal resolution resulting in 3 hourly average concentrations. The grid cell and time step matching each individual measurement were taken using a 4 dimensional bi-linear interpolation to the nearest model space and time coordinate. For the analysis we used three aggregated model species: TM, GEM, and OM = TM – GEM."

3) It seems to me that the manuscript was not thoroughly proof-read before publication in ACPD. There are several minor errors that often are distracting. For example, the citation 'Lyman and Jaffe, 2012', was cited at times as 'Lyman and Jaffe, 2011', 'Lyman and Jaffe et al., 2012', 'Lymann and Jaffe, 2012', and 'Jaffe and Lyman, 2012'). There are also a few instances where abbreviations were used without prior definitions. For example, in the main text RM was used first on Line 515, but not defined until Line 663, while in the abstract it was referred to as oxidized mercury, but in the main text as reactive mercury. I recommend that the manuscript be thoroughly proof-read before final submission.

A: I want to apologize for any inconveniences for the reviewer.
We corrected this and other errors in the citations and did a more thorough proof reading of the revised manuscript.

Specific comments:

1) Sect. 2.3: What was the spin-up period for the sensitivity simulations?

A: The specification for the MMTF scenario model runs was a spin up time of at least 2 years starting from the BASE case spin up. We included this information.

2) Lines 452-457: The underestimate in GOM concentrations seems to be related to the ambient absolute humidity and ozone concentrations and is likely not systematic, as stated by the authors.

A: We clarified this and added more statistical analyses to the results section.

"Generally, the model error can be separated into three parts: The bias, which represents any systematic errors, the variance which gives the variability around the mean value, and the covariance which represents the correlation between model and observations (Solazzo and Galmarini, 2016). By using MDPs we completely remove the bias and all systematic errors from our evaluation. Combining MDP and correlation coefficient, we are able to investigate the models capabilities to reproduce areas with high and low production of oxidized mercury and the influence of different chemistry schemes. The idea behind this is that even if the absolute measurements are not correct, we can use them to identify regions with mercury oxidation in the vertical column."

3) Line 543: Do the authors mean measured 'variability' instead of 'uncertainty'? Same for Lines 699, 797.

A: No we refer to the actual uncertainty of the observations as published in the ETMEP, CARIBIC, and NOMADSS datasets. For ETMEP the uncertainty is based on measurements from two co-located Tekran instruments on board of the air-craft for GEM and on denuder blank measurements for GOM. For NOMADSS the uncertainty is based on the lower limit of detection of the DOGHS instrument.

4) Line 567: It is stated that CCLM-CMAQ has the tropopause as its upper boundary, but Fig. 2 shows model values for CCLM-CMAQ up to 18 km.

A: This is only an artifact in the plot which we corrected.

5) Line 580: 'Linear TM'. This term is not clear.

A: We mean the constant TM concentrations (→ the missing trend) inside the free troposphere. We corrected this.

6) Lines 910-920: The authors interpret the high RM above 6 km as being related to stratospheric transport of Br and cite Gratz et al. (2015). However, Gratz et al. (2015) did not find evidence of stratospheric intrusion, and the authors' conclusions seem contradictory to that study. Is it possible to reconcile these two interpretations?

A: This is correct and we modified our conclusions accordingly.
We still think that stratospheric intrusions are an important source for Br in the upper troposphere during spring time. However, in the episode during July 2013 as described by Gratz et al. (2015) the water vapor concentrations seem too high to indicate a stratospheric origin of the air mass (the low ozone concentrations however could also be explained by depletion due to high Br concentrations).

"Our interpretation of the observations is that stratospheric intrusions and tropopause folds, which mainly occur during spring time, play an important role for elevated OM concentrations in the upper FT at altitudes above 6000m. The frequency of stratosphere to troposphere transport is regionally variable and has shown to be most common in the latitudes where the measurements were performed. However, also long range transport of marine bromine species as observed by Gratz et al. (2015) during the NOMADSS flights can be an important source of stratospheric Br. Thus, we emphasize the importance of further research regarding the atmospheric bromine cycle to better understand the oxidation pathways of mercury. Besides bromine species, stratosphere to troposphere transport could also be a source for OM already formed in the lower stratosphere. This could also explain the missing correlation of ozone concentrations and GEM/TM ratios measured by the CARIBIC aircraft in the upper FT."

7) Line 927: 'OH seems a plausible explanation'. How about O3? The models with O3 chemistry had better agreement with the observed concentrations for the NOMADSS profile.

A: This is correct, we adjusted the paragraph accordingly. Moreover, we included the calculated correlation coefficients into the discussion.

8) Lines 940-942: I am not sure how the higher effective height of emissions would affect GEM concentrations in the upper troposphere.

A: Emissions to high altitudes, especially when the effective emission height is above the PBL, have a longer atmospheric lifetime. On average, it will simply take longer for the substance to collide with a surface and undergo dry deposition. Moreover, is will not be scavenged by low altitude precipitation. This is only a hypothesis, but has been shown to be true for other pollutants e.g. SO4 emissions from coal fired power plants (Bieser et al., 2011) which we think is a good proxy for Hg emissions from the same source.

*Bieser J, Aulinger A, Matthias V, Quante M, Denier van der Gon HAC. 2011. Vertical emission profiles for Europe based on plume rise calculations. Environ Pollut 159: 2935–2946. doi: 10.1016/j.envpol.2011.04.030*

9) Table 1: Is OH an aqueous phase oxidant of GEM in the GEOS-Chem model?

A: Yes, OH is only used in the aqueous phase. We improved Table 1 so it is easier to read.

10) Table 2: Since the results of the sensitivity simulations were not available for all models, I suggest adding a column specifying which models participated in each sensitivity run.

A: We added this to Table 2.

11) Title: Perhaps the title should contain "Vertical and inter-hemispheric distribution of mercury species".

A: This is a very good suggestion and we changed the title accordingly.

Technical comments:

A: Thanks for the detailed list of minor errors. We corrected these in the revised version of the manuscript.

1) Lines 83-84: Grammar

2) Lines 98-102: Grammar

3) Line 143: Spelling, 'Woll'

4) Line 144: What does DOHGS stand for?
University of Washington Detector for Oxidized Hg Species

5) Line 620: '14th June' seems to be typo.

6) Line 665: Do the authors mean equilibrium between GOM and PBM?
yes

7) Table 1: No emission speciation information for WRF-Chem

8) Table 1: No reference for natural emissions for the first six models.

9) Table 1: Missing footnote 'a'.
we corrected table 1

10) Table 2: Refers to the emission set as UNEP2010, while Table 1 refers to it as AMAP.
Both should be AMAP/UNEP

11) Fig. 3: Lower panel. Is MDP for TM or GEM?
Fig 3 to 6 give MDP for GEM only.

12) Fig.7-9: Is Hg2+ different from RM?
Hg2+ is identical to oxidized mercury (OM) We think this is correct as we now use the term oxidized mercury instead of reactive mercury (RM).

13) Figs. 8-9: Units for Hg2+.
Corrected to pg/m³

14) Fig. 8: Should the x-axis label be MDP instead of PMB?
yes

---

## Author Comment (AC3) · 11 Apr 2017

Answers to Reviewer #2:

We want to thank reviewer #2 for pointing out weaknesses of the presented manuscript. We improved our terminology and added a section on the total mercury burden in the atmosphere to address this review.

1) Please clarify whether the heights mentioned throughout the manuscript are referred to the height above sea level (asl) or above ground level (agl). If the height refers to als, the authors should also compare their modeling results with observations at high-altitude peaks worldwide (e.g., Mount Bachelor Observatory, USA, 2700 m asl; Storm Peak, USA< 3200 m asl; Lulin Atmospheric Background Station, Taiwan, 2862 m asl; Pic du Midi Observatory, France, 2877 m asl and Mt. Leigong, China, 2178 m asl. These observations are in the free troposphere and can be compared with the modeling results).

A: The hight levels refer to agl (above ground level) as all models use sigma-hybrid levels for the vertical coordinate. This makes it difficult to compare the model results to mountain stations. This is especially true for the global models which use quite low horizontal resolutions. Thus, we did not compare modeled concentrations against observations from mountain stations.

2) Line 480: Please clarify the mean of 'source regions'. Are they related to anthropogenic or natural sources (GOM and PBM formation in the atmosphere)?

A: We clarified this: "… even in source regions with high anthropogenic emissions (e.g. coal fired power plants)."

3) Line 480: the citation should be Fu et al., 2016.

A: We added: "In China, PBM concentrations up to 1000 pg/m³ and GOM concentrations up to 100 pg/m³ have been observed, however no aircraft observations in the PBL and the lower free troposphere are available for this region. (Fu et al., 2016)."

4) The authors modeled the vertical concentrations of GEM, GOM, and PBM in the troposphere and stratosphere. Will it be possible to give the total quantity (Mg) of GEM, GOM, and PBM in the PBL, lower free troposphere, middle free troposphere, upper free troposphere and stratosphere?

A: We added a short section investigating the total atmospheric mercury burden as calculated by the four global models:

*3.4 Total atmospheric mercury burden*

*We investigated the total atmospheric mercury burden as predicted by the four global models. We found that all models give a similar relative global mercury distribution with 53% to 55% of the TM in the northern hemisphere. Looking at the vertical distribution the models predict 22% to 34% inside the PBL, 54% to 60% in the free troposphere, and 9% to 16% in the stratosphere. However, the absolute numbers show a large variability. ECHMERIT (1800 Mg) gives the lowest total atmospheric mercury burden, followed by GEOS-Chem (3700 Mg), GLEMOS (6200 Mg) and GEM-MACH-Hg (6300 Mg) (Fig. 15). On average the models give 4500 Mg which is close to the estimate of 5300 Mg by Amos et al. (2013). The average vertical distribution in the model ensemble is PBL (1300 Mg), FT (2600 Mg), and stratosphere (600 Mg).*

[Figure]

Figure 15: Global cumulative total mercury (solid) and gaseous elemental mercury (dashed line) integrated from surface to model level for each of the four global models. The model ensemble gives a total 4500 Mg of mercury in the atmosphere with 1300 Mg inside the PBL, 2600 Mg in the free troposphere, and 600 Mg in the stratosphere.

5) The atmospheric physicochemical properties over the oceans and continents are generally different. I suggest the authors should also calculate the average vertical distributions of atmospheric Hg species over oceans and continents.

A: This is a interesting idea, however we find that the paper is already extremely long and thus did not add this to the revised manuscript.

6) The measurements of GOM and PBM have many uncertainties. As mentioned the in the paper, previous studies for GOM and PBM measured utilized several different techniques. The authors should introduce the uncertainties of these observations and the effects on their comparisons.

A: The observations used for this evaluation are all based on Tekran instruments. Thus, all observations are comparable to each other. We added more discussion on the representativeness of the data and the impact of ozone and humidity for the retrieval of oxidized mercury species by the Tekran instruments.
(please see also answers to reviewer #1)

7) Line 1418, Shah et al., 2015 should be Shah et al., 2016.

A: we corrected this typo throughout the manuscript.

8) GEM, GOM, and PBM (generally the Hg bounded with fine particulates) are the three major forms of atmospheric Hg. In many parts of the paper, the authors used TM (total atmospheric Hg) and RM (reactive atmospheric Hg), and this is not completely right under some situations. Hg bounded with coarse particulates could represent a large fraction of total particulate Hg in the PBL. Also, GEM, GOM, and PBM could be transformed to other Hg species including Hg in cloud vapor, fog, etc.. These Hg species in the atmosphere sometimes represents an important fraction of atmospheric Hg. Should we define these species as RM? Have the authors taken these species into the modeling? I think this might be an important element influencing the comparisons between observations and modeling

A: We agree with the reviewer that the name RM is misleading and not the correct term to use. We now use OM (for oxidized mercury) instead throughout the manuscript.
In the manuscript, OM is defined as the sum of all oxidized mercury species including the aqueous phase. Thus, OM = TM – GEM

At the end of the introdcution we now state:
"The speciation of mercury is thus operationally defined as GEM, GOM, and PBM (Gustin et al., 2015). In the following we will address the sum of all oxidized mercury species, including mercury in the aqueous phase, as OM (oxidized mercury)."